# Single-cell dynamics of chromatin activity during cell lineage differentiation in *Caenorhabditis elegans* embryos

Zhiguang Zhao[1,2,†] (iD), Rong Fan[1,2,†], Weina Xu[1,2] (iD), Yahui Kou[1,2], Yangyang Wang[1], Xuehua Ma[1] & Zhuo Du[1,2,*] (iD)

## Abstract

Elucidating the chromatin dynamics that orchestrate embryogenesis is a fundamental question in developmental biology. Here, we exploit position effects on expression as an indicator of chromatin activity and infer the chromatin activity landscape in every lineaged cell during *Caenorhabditis elegans* early embryogenesis. Systems-level analyses reveal that chromatin activity distinguishes cellular states and correlates with fate patterning in the early embryos. As cell lineage unfolds, chromatin activity diversifies in a lineage-dependent manner, with switch-like changes accompanying anterior–posterior fate asymmetry and characteristic landscapes being established in different cell lineages. Upon tissue differentiation, cellular chromatin from distinct lineages converges according to tissue types but retains stable memories of lineage history, contributing to intra-tissue cell heterogeneity. However, the chromatin landscapes of cells organized in a left–right symmetric pattern are predetermined to be analogous in early progenitors so as to pre-set equivalent states. Finally, genome-wide analysis identifies many regions exhibiting concordant chromatin activity changes that mediate the co-regulation of functionally related genes during differentiation. Collectively, our study reveals the developmental and genomic dynamics of chromatin activity at the single-cell level.

**Keywords** *Caenorhabditis elegans*; chromatin activity; fate patterning; position effect; single-cell lineage tracing

**Subject Categories** Chromatin, Transcription & Genomics; Development

**Mol Syst Biol. (2021) 17: e10075**

## Introduction

Chromatin state regulates diverse biological processes by controlling gene expression potential. During *in vivo* development, specific cells with identical genotypes differentiate into different cell types through the spatiotemporal expression of distinct sets of genes (Yadav *et al*, 2018; Packer *et al*, 2019). Elucidating how dynamic and cell-specific chromatin states regulate *in vivo* cell differentiation processes has been a fundamental question in the fields of epigenetics and developmental biology (Yadav *et al*, 2018). Accurate assessments of chromatin state are essential to understanding how it functions in development. A combination of molecular approaches and high-throughput sequencing has been widely used to analyze the biochemical and biophysical properties of chromatin, including histone modifications, chromatin accessibility, and the spatial organization of chromatin (Goodwin *et al*, 2016). These epigenomic approaches have significantly enhanced the molecular profiling and mechanistic analysis of chromatin regulation during cell differentiation (Nicetto *et al*, 2019).

In addition to the epigenomic approaches described above, another powerful strategy for elucidating the functional state of chromatin is position-effect variegation (Schotta *et al*, 2003; Elgin & Reuter, 2013), which is a classic phenotype associated with *Drosophila* eye color. The white-eyed mutant phenotype is caused by the epigenetic silencing of normally active white genes, which is due to misplacement in heterochromatin regions (Timms *et al*, 2016). These findings established that chromatin environments exhibit strong positional effects on modulating the expression potential of nearby genes. The eye color phenotype provides a straightforward readout of the functional state of chromatin; thus, it has been widely used in genetic screens to identify potential regulators of chromatin activity. These efforts have identified many critical chromatin regulators that constitute much of our current knowledge of chromatin biology, including heterochromatin protein 1 (HP1) (James & Elgin, 1986), tri-methylation of histone H3 at lysine 9 (Rea *et al*, 2000), and histone deacetylase (Mottus *et al*, 2000). Furthermore, the position effects on reporter gene expression have also been used to identify novel chromatin regulators in higher organisms, such as the human silencing hub complex HUSH (Blewitt *et al*, 2005; Ashe *et al*, 2008; Tchasovnikarova *et al*, 2015). In addition to identifying chromatin regulators, position effects on gene expression have also been

1   State Key Laboratory of Molecular Developmental Biology, Institute of Genetics and Developmental Biology, Chinese Academy of Sciences, Beijing, China
2   University of Chinese Academy of Sciences, Beijing, China
   *Corresponding author. Tel: +86 10 64801699; E-mail: zdu@genetics.ac.cn
   †These authors contributed equally to this work

exploited to elucidate the chromatin activity across the genome. In these studies, the expression status of the same reporter gene integrated into dozens to thousands of different genomic positions has been used as a chromatin activity sensor (Gierman *et al,* 2007; Akhtar *et al,* 2013; Chen *et al,* 2013; Frokjaer-Jensen *et al,* 2016). The analysis of position effects across the genome has provided insight into several aspects of how the chromatin activity is organized throughout the genome and how it is regulated, including chromatin domain organization (Gierman *et al,* 2007; Akhtar *et al,* 2013), the structural properties of chromatin during the regulation of gene expression (Gierman *et al,* 2007), the repressive role of the lamina during gene expression (Akhtar *et al,* 2013), and germline gene silencing (Frokjaer-Jensen *et al,* 2016).

As compared to existing sequencing-based epigenomic approaches, the assessment of position effects represents a unique approach that can be used to measure the functional state of chromatin as it measures ultimate gene expression levels, the outcomes of chromatin state. However, applying this approach to developing single cells during cell differentiation remains challenging. Previous large-scale analyses of position effects primarily focused on dissecting the genomic properties and regulation of chromatin activity. Thus, these studies involved single-cell organisms, cell lines, and measuring reporter gene expression in multicellular organisms at the tissue/organism level. A systematic analysis of position effects associated with single cells during *in vivo* differentiation has not been previously performed.

In this work, we exploited the position effects in single cells to elucidate the chromatin activity landscape during embryogenesis of *Caenorhabditis elegans*, a widely used multicellular model organism for studying developmental regulation at the single-cell level. Using a live-imaging approach, we quantified the expression levels of a reporter gene that was integrated into more than 100 positions throughout the genome, and the resulting data were used to infer chromatin activity landscapes (changes in reporter expression levels across genomic positions) corresponding to all lineaged single cells during *C. elegans* early embryogenesis. We revealed the general dynamic patterns of chromatin activity accompanying critical processes of *in vivo* cell lineage differentiation, including lineage commitment, anterior–posterior fate asymmetry, tissue differentiation, cell heterogeneity, and bilateral symmetry establishment. Our

findings contribute to a systems-level understanding of the developmental and genomic dynamics of chromatin activity at the single-cell level.

## Results

### Quantification of the position effects on reporter gene expression in lineage-resolved single cells

A collection of transgenic *C. elegans* strains (integrants) has been previously generated (Frokjaer-Jensen *et al,* 2014), each containing a single copy of the same GFP-expressing cassette driven by a ubiquitous promoter (*eef-1A.1*, a translational elongation factor) and integrated throughout the genome (Fig 1A and Dataset EV1). Imaging representative integrants revealed strong position- and cell-dependent variation of GFP expression (Fig 1B), which suggests that P*eef-1A.1*::GFP is highly responsive to different chromatin environments that environment exhibits cell specificity.

To allow single-cell GFP quantification, another nucleus-localized, ubiquitously expressed mCherry transgene was crossed into the above strains for nuclei identification and tracing (Dataset EV1). We performed 3D time-lapse imaging to record embryogenesis at high spatiotemporal resolution, and images were analyzed to determine GFP expression in individual cells (Fig 1C) (Murray *et al,* 2008; Du *et al,* 2014). Cell lineages were reconstructed based on the automatic identification and tracing of all cells via the mCherry signal using StarryNite and AceTree software (Bao *et al,* 2006; Boyle *et al,* 2006; Santella *et al,* 2010; Santella *et al,* 2014; Katzman *et al,* 2018), followed by multiple rounds of manual curations (Fig 1D). Simultaneously, the intensity of GFP in each traced nucleus at each time point was measured and averaged to indicate chromatin activity at a certain genomic position (Fig 1D and E). On average, GFP expression was measured at 38 consecutive time points (range 13–83) for each traced cell (Figs 1D and EV1A). In total, we measured GFP expression in 268 embryos to quantify chromatin activity at 113 genomic positions (at 0.88-Mb resolution) in 722 lineaged cells (364 traced terminal cells), which covers all cells up to the 350-cell stage, at which tissue fate specification is completed (Appendix Fig S1 and Dataset EV2).

---

**Figure 1.  Construction of chromatin activity landscapes in lineage-resolved single cells.**

A   Transgenic strains carrying the same P*eef-1A.1*::GFP::NLS expression cassette integrated into different genomic positions (green boxes) across the genome. NLS denotes the nuclear localization signal.

B   Representative micrographs showing cellular expression of ubiquitous mCherry (left) and P*eef-1A.1*::GFP (right) when integrated into different genomic positions. Scale bar = 10 μm.

C   3D time-lapse imaging of *Caenorhabditis elegans* embryogenesis. 3D maximum projections show cellular expression of GFP and ubiquitous mCherry (lineage marker) at five characteristic developmental stages. Scale bar = 10 μm.

D   Schematic of automated cell identification (circles), lineage tracing (arrows with bifurcations indicating cell divisions), and continuous quantification of P*eef-1A.1*::GFP expression levels (green).

E   Quantification of P*eef-1A.1*::GFP expression in every lineaged cell. The color-coded tree visualizes the single-cell expression of GFP (green) integrated into a position in all lineaged cells. Vertical lines indicate cells traced over time, and horizontal lines indicate cell divisions.

F   Construction of chromatin activity landscape. Left: an example illustrating the inference of chromatin activity landscape by integrating P*eef-1A.1*::GFP expression levels across four genomic positions in an equivalent cell (black circle, ABplpappaa). Right: Circos plot showing the cellular chromatin activity landscape, with the radial line indicating the genomic location and color gradient indicating GFP expression level.

G   Images showing expression of ubiquitous mCherry (top) and P*eef-1A.1*::GFP (bottom) in the ABplpappaa cell at all integration sites on chromosome I. Scale bar = 1 μm.

H   Circos plots showing chromatin activity landscapes of nine representative embryonic cells with color gradient indicating chromatin activity level.

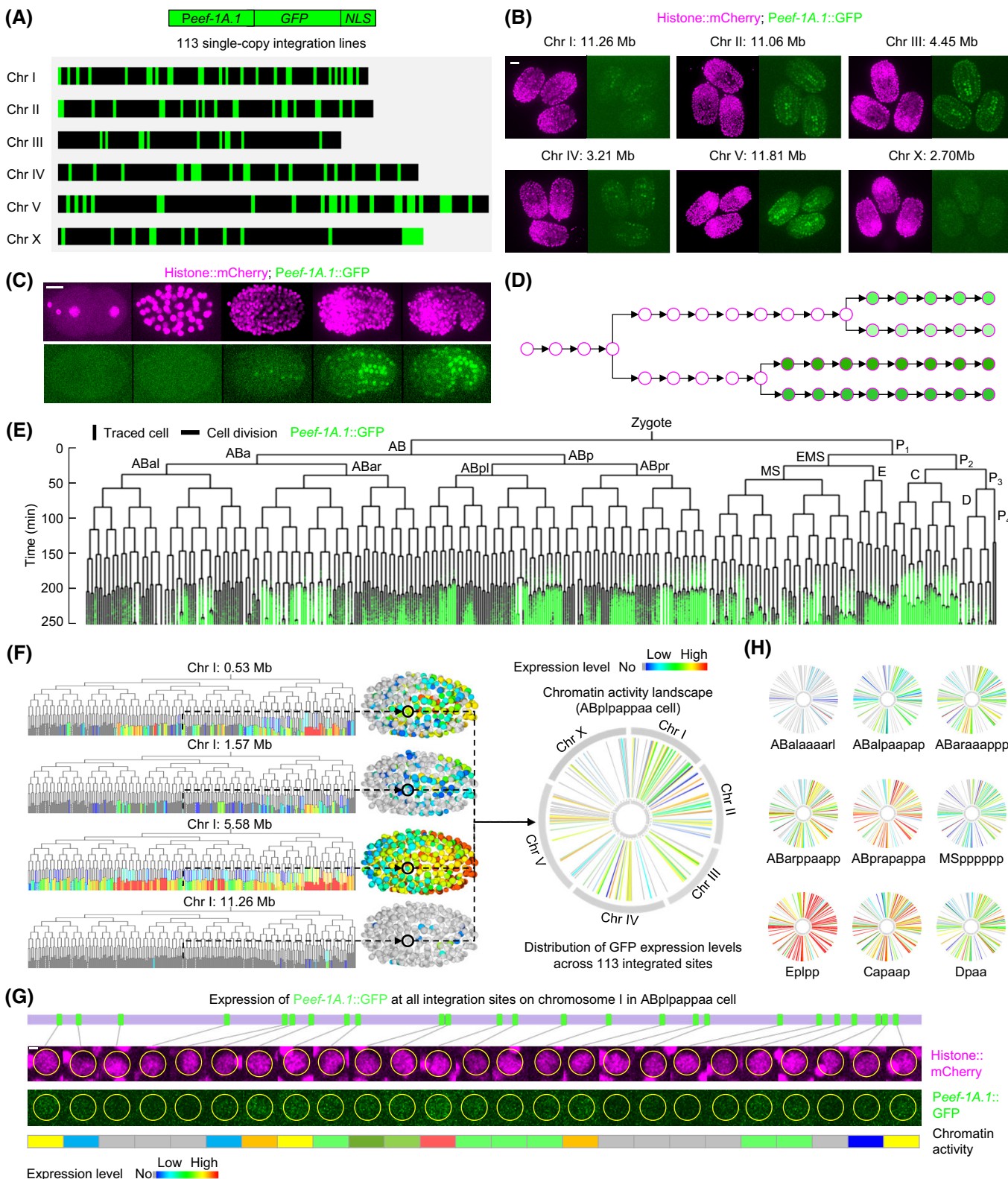

Figure 1.

We validated the accuracy of the annotation of GFP integration sites that had been done previously (Fig EV1B) and the reliability of the lineage tracing results (Materials and Methods). Additionally, we optimized the expression quantification method to compensate for the attenuation of fluorescence intensity with sample depth (Fig EV1C–F) and to precisely measure instantaneous GFP

expression in a single cell (Materials and Methods). These improvements allowed for a more reliable comparison of reporter expression across cells. Cellular GFP expression was reproducible, with a median correlation coefficient of 0.83 and a median consistency of binarized expression of 0.90 between experimental replicates (Fig EV1G).

*Caenorhabditis elegans* embryogenesis follows a fixed cell lineage pattern and generates the same set of differentiated cells in each embryo, making embryos entirely comparable at single-cell resolution (Sulston *et al*, 1983). Thus, although the cellular GFP expression at each integration site was assayed in individual embryos (Appendix Fig S1), the invariant lineage allowed a cell-by-cell integration of GFP expression levels across integrants in lineage-equivalent cells (Fig 1F). Using GFP expression levels at different genomic positions as a sensor of chromatin activity, we constructed the distribution of chromatin activity across 113 genomic positions (termed the chromatin activity landscape) for all lineage-traced cells (Fig 1F–H and Dataset EV3). For multiple replicates of GFP expression for the same integrant (range 2–8), only those expressed in more than 60% of replicates were considered as being expressed, and levels were averaged to represent the consensus chromatin activity. Aggregating GFP expression across all positions showed that P*eef-1A.1*::GFP was predominantly expressed in the majority of cells from the 350-cell stage (with 364 cells) and onward (Fig EV1H), making the analysis of chromatin activity in cells from this stage the most informative. Unless otherwise specified, all analyses focused on the 364 traced terminal cells.

## Position effects on GFP expression reliably indicate chromatin activity

We systematically validated the biological relevance of the inferred chromatin activity landscape by determining whether the measured position effects on P*eef-1A.1*::GFP expression are concordant with chromatin features related to its activity in regulating gene expression (Fig EV2 and Dataset EV4).

First, position effects on GFP levels were consistent with the known genomic distribution of chromatin activity. Similar to a previous finding (Frokjaer-Jensen *et al*, 2016), the average GFP levels across cells were significantly higher when integrated into autosomes as compared to the X chromosome and were significantly higher when integrated into the central region of the chromosome as compared to the arm region (Fig EV2A and B).

Second, the average GFP levels were consistent with the biochemical properties of chromatin that are related to its activity. We examined the correlation between GFP expression and histone modification, a key determinant of chromatin activity. Prediction of GFP levels at each integration site using the combination of 19 types of histone modifications (Ho *et al*, 2014) revealed that the predicted values correlated significantly with the measured ones (Fig EV2C; $R = 0.82$, $P = 8.27E-29$). Twelve histone modifications correlated significantly with GFP levels (Fig EV2D), and perturbing the writers of representative repressive (tri-methylation of histone H3 at lysine 9, H3K9me3) and activating modifications (acetylation of histone H4 at lysine 16, H4K16ac) induced expected changes in GFP expression at multiple integration sites (Fig EV2E and F). These results confirm that GFP expression is both indicative of and responsive to chromatin activity. Previous studies have also segregated chromatin

into various states/domains exhibiting differential activity (Ho *et al*, 2014; Evans *et al*, 2016). Using the embryonic dataset, we found that average GFP expression levels were significantly higher when located in active chromatin domain/states than when located in silent regions (Fig EV2G and H).

Third, GFP expression was concordant with the biophysical properties of chromatin that influence its activity. Using previously generated accessible chromatin by ATAC-seq at several developmental stages (Janes *et al*, 2018), we found that the average GFP levels were significantly higher when integrated into accessible chromatin regions than the rest of the genome (Fig EV2I). Moreover, GFP expression was consistent with the 3D localization of chromatin. Chromatin regions attached to the nuclear lamina tend to be heterochromatic and exhibit low transcriptional activity (Reddy *et al*, 2008; Kind *et al*, 2013). We examined whether GFP expression was reduced when located in the lamina-associated domain (LAD). LAD information was obtained from a previous study that determined regions associated with LEM-2, an inner nuclear membrane protein, in mixed-stage *C. elegans* embryos (Ikegami *et al*, 2010). Indeed, the average GFP levels were significantly lower when the transgene was located in the LAD as compared to non-LAD regions (Fig EV2J).

Conversely, GFP expression was not associated with local genetic environments. Integrants located in intragenic and intergenic regions exhibited comparable expression levels (Fig EV2K). Furthermore, no correlation between the average expression level of GFP and endogenous genes located in a 5-kb interval centered on the integration sites was observed (Fig EV2L, left, $R = 0.13$, $P = 0.24$), indicating that local genetic environments (*cis*-elements of endogenous genes) did not significantly influence P*eef-1A.1*::GFP expression. This result is consistent with a previous yeast study in which the position effects on the expression of a $kan^R$ gene driven by the *TEF* promoter do not correlate with the expression of endogenous genes at the same position (Chen *et al*, 2013). We observed a modest but significant correlation when interval size was extended to over 100 kb, with the 500-kb interval yielding the strongest correlation (Fig EV2L, right, $R = 0.57$, $P = 3.76E-11$). Since gene expression over a large chromosome domain is more likely to be governed by chromatin as the influence of *cis*-elements on gene expression would be partially normalized, this result again suggests that the position effects on GFP expression are due to differential chromatin activity.

Single-cell analysis also supported the reliability of the inferred chromatin activity landscape. Using endogenous gene expression over a large genomic interval as proxy for chromatin activity, we compared GFP expression to single-cell RNA-sequencing (scRNA-seq) data (Packer *et al*, 2019). The result showed that cellular GFP expression across all positions was concordant with endogenous gene expression in the same cells at the 500-kb interval (Fig EV2M and Dataset EV5), supporting that the landscapes determined from position effects on GFP expression in single cells also indicate chromatin activity. Due to the scarcity of single-cell chromatin data, we were unable to assess the relevance of cellular chromatin activity in the context of other chromatin properties.

Collectively, systematic comparisons of GFP expression to the genomic distribution, biochemical, and biophysical properties of chromatin support that the inferred chromatin landscape using position effects on P*eef-1A.1*::GFP indicates chromatin activity.

## Cellular chromatin activity is dynamic and informative for distinguishing cellular states

Having established that the inferred chromatin activity is reliable, we next determined the extent to which the activity changes across positions and cells (Fig 2A). We first examined whether the expression of P*eef-1A.1*::GFP changes considerably with genomic position, taking the expression variability at each position into account (Fig 2 B). For both quantitative expression and binarized expression, analysis revealed that the Pearson correlation coefficient (*R*) of GFP expression between different integration sites was significantly lower than that between replicates at a given position (Fig 2C and D). Pair-wise comparisons likewise showed that the cellular pattern of GFP expression at a given integration site was, on average, distinct (*R* < 0.5) from 40% (quantitative expression) and 92% (binarized expression) of the patterns resulting from other integration sites. These results suggest that the *eef-1A.1* promoter sequence does not significantly dominate the position effects on GFP expression. It should be noted that the promoter used in this study is a well-known strong promoter, which might account for quantitative reduction in expression being more frequently observed than on/off changes. Thus, a considerable quantitative reduction in chromatin activity assayed here could correspond to more dramatic (on/off) changes in many endogenous contexts.

While GFP expression was generally consistent between experiment replicates (Fig EV1G), highly variable expression was observed in certain cells at certain genomic positions (Appendix Fig S2). Because only a small number of experimental replicates were performed, we selected four insertion strains exhibiting high variability and quantified GFP expression in more embryos. The correlation of GFP expression between replicates at these positions

remained low (Appendix Fig S2), suggesting chromatin activity in certain regions could be flexible. Indeed, chromatin state/activity has been previously shown to be variable and stochastic at certain positions (Angermueller *et al*, 2016). Furthermore, when studying the position-effect variegation phenotype of fly eye color, epigenetic silencing of the *white* gene has been shown to be highly stochastic, causing a variegated red and white color phenotype (Timms *et al*, 2016).

Qualitatively, chromatin activity is highly dynamic across the genome and among cells. This was determined by comparing cellular and positional dynamics of the on/off status of GFP expression. At each integration site, GFP was expressed in a proportion of the 364 cells (Fig 2E, median = 60%); at only seven positions, GFP was active (*n* = 1) or silenced (*n* = 6) in all cells. It suggests that chromatin activity at most genomic positions exhibited cell specificity. In each cell, GFP was expressed only when it had been integrated into a fraction of genomic positions (Fig 2F, median = 67%), suggesting that chromatin activity in a cell is positionally specific. In only a small number of cells, the GFP was constitutively expressed or not expressed across all integration sites. Specifically, in < 10% of the cells (*n* = 35), GFP was constitutively expressed (expressed at > 80% of the integration sites), and in < 5% of the cells (*n* = 15), GFP was constitutively silenced (expressed at < 20% of the integration sites). It suggests that when changing the chromosome location, *eef-1A.1* promoter in a cell could either be on or off, and that the promoter does not exhibit strong expression bias to many cells.

We then examined to what extent the cellular chromatin activity landscape can distinguish individual cells. We compared the GFP expression patterns across cells and calculated for each pair-wise comparison the number of integration sites at which the GFP

---

**Figure 2. Cellular chromatin activity landscape is informative for distinguishing cells.**

A   Heatmap showing GFP expression levels in individual cells (*n* = 364, ordered by lineage origin) when integrated into different genomic positions (*n* = 113, clustered based on expression pattern).

B   Quantification of GFP expression similarity between integrants. Pearson correlation coefficient is measured to quantify expression similarity across all cells between experimental replicates at the same integration site and between different integration sites.

C, D   Distribution of GFP expression similarity between experimental replicates and between different integrants, calculated using quantitative (C) or binarized (D) expression levels. Violin plot: The center white point is the median, box limits are the first and third quartiles, box length indicates interquartile range (IQR), and whiskers either 1.5 times the IQR or the minimum/maximum value if it falls within a factor of 1.5 times of the IQR. Violin plot bandwidth is estimated by "scott" method.

E   Distribution of the proportion of cells that express GFP at each integration site. Schematics on the top show expressing cells (green) of representative integrants.

F   Distribution of the proportion of integration sites at which GFP is expressed in each cell. Schematics on the top show the expressed integrants (green) in representative cells.

G   Heatmap showing the number of integration sites at which the on/off expression status of GFP is distinct between each pair-wise cell comparison. Integration sites that exhibit variable expression status between experimental replicates were not considered. Cells are ordered according to tissue types (Neu, neuronal system; Pha, pharynx; Ski, skin; Mus, body wall muscle; Int, intestine), with those that do not belong to a specific tissue type omitted.

H   Box plots show the fraction of cells of each tissue type whose chromatin activity landscapes are distinct compared with all other cells belonging to the same or different tissue types (cell numbers: *n* = 128 for Neu, 44 for Pha, 75 for Ski, 44 for Mus, and 16 for Int). The center band of the box is the median, box limits are the first and third quartiles, box length indicates IQR, and whiskers either 1.5 times the IQR or the minimum/maximum value if it falls within a factor of 1.5 times of the IQR. Outliers not shown. A cell is defined as having a distinct chromatin activity landscape if GFP expression at five or more of the 113 assayed genomic positions exhibits distinct on/off expression status compared with another cell.

I   Heatmap showing GFP expression levels in individual cells (*n* = 364) when integrated into previously defined genomic regions having global silent or active chromatin state (Ho *et al*, 2014). On the right are indicated four representative integration sites at which GFP expression is distinct from the global pattern.

J   Distribution of Pearson correlation coefficients comparing the observed chromatin activity landscapes in individual cells to those predicted by a combination of 19 types of histone modifications (*n* = 364).

K   Correlation of cellular chromatin activity landscape to the global landscape predicted by histone modifications, contextualized by cell lineage origin and tissue fate. On the lineage tree, traced terminal cells are color-coded according to the correlation coefficient between observed and predicted chromatin activity; the bottom 25% (blue, *n* = 91) are considered divergent. The barcode below the cell lineage indicates the tissue fate of each cell.

Source data are available online for this figure.

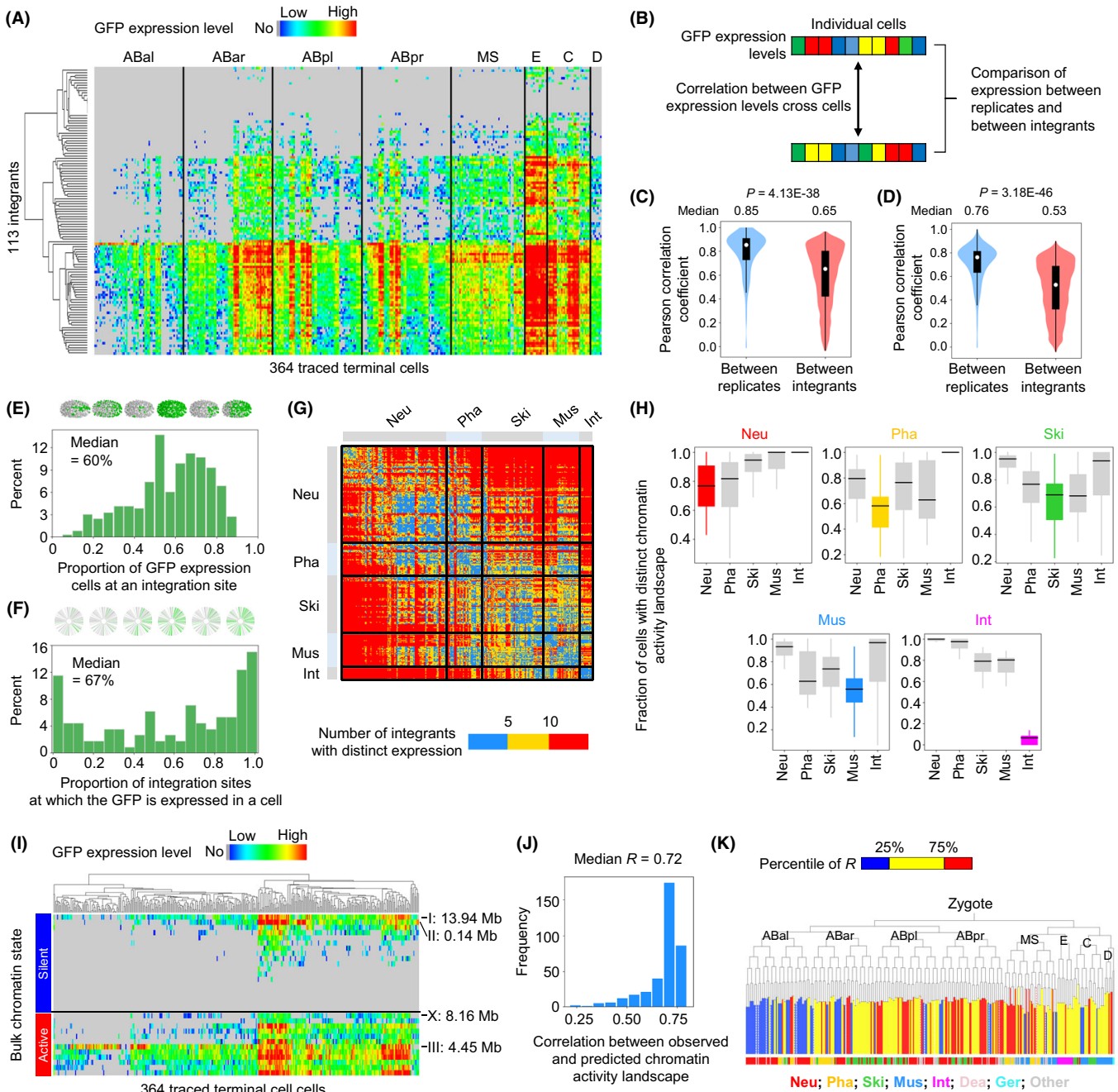

Figure 2.

expression status is distinct. Intriguingly, for most cell–cell comparisons, the binary GFP expression at many integration sites was distinct, and at a considerable number of integration sites, the expression status can distinguish a cell from many other cells (Fig 2G). On average, cellular chromatin activity distinguished a cell from 79.4% of other cells if the on/off state of chromatin activity at five or more genomic positions was distinct in a cell–cell comparison. The tissue-level analysis further showed that chromatin activity landscape not only distinguished a cell from a large fraction of cells belonging to a different tissue type but also distinguished it from a considerable fraction of cells of the same tissue type (Fig 2H). The only exception was the intestine cells, for which distinguishing chromatin activity

occurred at only a few integration sites. One possibility is that intestine cells exhibit limited functional diversification due to being derived clonally from a single progenitor cell (called E) and all intestine cells or their progenitors uniformly expressing most master regulators of intestine differentiation (Maduro & Rothman, 2002; McGhee, 2007). Thus, in general, the chromatin activity landscape provides rich information for distinguishing cells at a sub-tissue level.

Finally, the cellular resolution of our data enabled identifying potential cell- and position-specific chromatin activity that is unable to be obtained from the bulk data of cell populations. Although the chromatin activity described here is globally concordant with previously defined chromatin regions for which a silent/active state is

evident in bulk epigenomic data (Fig EV2G and H), this activity exhibited considerable cell specificity at given genomic positions. For example, certain integrants located in regions with a global silent/active state exhibited divergent activity in specific cells (Fig 2 I). Moreover, certain cells also exhibited chromatin activity landscapes that diverged from those predicted by cell population-based histone modification datasets (Ho *et al*, 2014). Correlation analysis comparing the observed chromatin activity landscape in each cell with that predicted by a combination of 19 types of histone modifications revealed that, in certain cells, the predictive power of bulk histone modifications was low (Fig 2J). Interestingly, these cells (with an *R* ranked in the bottom 25%) were significantly enriched for neuronal cells from the ABal lineage (Fig 2K, 3.36-fold enrichment, *P* = 4.22E-21, Fisher's exact test). Thus, a lineage-resolved single-cell dataset provides the opportunity to pinpoint genomic regions and cells having distinct chromatin activity.

All told, we generated a lineage-resolved chromatin activity landscape that is dynamic, informative, and biologically relevant in indicating the functional state of chromatin and in distinguishing cellular states. Taking advantage of this cellular landscape and the clarity of *C. elegans* cell annotations, we systematically investigate in the following sections the dynamics and potential implications of chromatin activity during cell lineage differentiation, including lineage specification, anterior–posterior fate asymmetry, tissue differentiation, cell heterogeneity, and bilateral symmetry establishment.

## Chromatin activity dynamics correlate with lineage-coupled cell differentiation

Cell differentiation accompanies lineage progression. The lineage-based mechanism plays a crucial role in initiating cell differentiation by assigning distinct fates to progenitor cells in a lineage-dependent manner, hence diversifying cell fates (Labouesse & Mango, 1999). Accordingly, it is natural to ask whether the cellular chromatin activity landscape diversifies during lineage progression and indicates lineage-coupled fate differentiation. We first quantified chromatin activity divergence as a function of the lineage relationship between 364 traced terminal cells. The divergence was measured as the Euclidean distance between chromatin activities (GFP expression) across all integration sites in a cell (Fig 3A). Lineage relationship was quantified as cell lineage distance, which was defined as the total number of cell divisions separating cells from their lowest common ancestor (Fig EV3A). In the majority of cases, higher divergences were observed between cells with a large lineage distance and, globally, the divergence increased progressively with cell lineage distance (Fig 3B and C). Thus, in general, chromatin activity landscape diversifies gradually across cells during lineage progression.

We next analyzed the lineage-coupled kinetics of cell differentiation by measuring how cell fates change as a function of the lineage distance between cells. Based on the lineage tree structure and tissue types of all terminally differentiated cells, we retrospectively defined progenitor cell fate as the combinatorial pattern of tissue types produced by each cell and quantified the fate difference between cells (Materials and Methods and Fig EV3B). This analysis showed, generally, as the cell lineage unfolds, cells differentiate progressively. The fate divergences between cells were proportional to their lineage distances at different developmental stages, similar to what was observed with chromatin activity divergences (Figs 3C and EV3C). To further demonstrate that chromatin activity dynamics were associated with fate changes, we directly analyzed the relationship between the two using cells with identical lineage

**Figure 3. Chromatin activity dynamics correlate with lineage-coupled fate differentiation.**

A    Comparison of chromatin activity divergence between two cells by measuring the Euclidian distance between GFP expression levels at all integration sites.

B    Heatmap showing the mean chromatin activity divergence (color gradient) between each of the 364 traced terminal cells (rows) to all other cells at different cell lineage distances (columns).

C    Changes in chromatin (blue) and fate (red) divergences (mean ± SD) following an increase in cell lineage distance of paired cells (cell pair numbers from left to right: $n$ = 179; 341; 666; 1,319; 2,530; 4,680; 8,522; 16,384; 10,752). Because an odd-numbered lineage distance involves cells at different generations, only those with an even number (80% of the cases) were used to analyze the relationship between cell lineage distance and other cellular attributes.

D    Each panel compares changes in fate divergences (mean ± 95% CI) between cells that exhibit different chromatin activity divergence levels (divided into three bins, cell pair numbers within each bin, from left to right: $n$ = 408; 174; 84 for $D$ = 6; $n$ = 265; 902; 152 for $D$ = 8; $n$ = 400; 1,798; 332 for $D$ = 10; $n$ = 2,813; 1,420; 447 for $D$ = 12; $n$ = 894; 4,907; 2,721 for $D$ = 14) for cell pairs having an identical lineage distance ($D$). Statistics: Mann–Whitney $U$-test. ***$P$ < 0.001.

E    Tree visualization of the inferred chromatin activity transition points (green dots) and associated fate divergence (color-coded vertical lines) between two daughter cells of each early progenitor cells. Fate divergence (ranging from 0 to 1) was evenly divided into four categories.

F, G    Two examples showing the association between chromatin activity transition and anterior–posterior fate asymmetry following EMS (F) and ABprapp (G) cell divisions. Left: comparison of intra- and inter-daughter–lineage chromatin activity divergences following a cell division (Intra-1, Intra-2 and inter-lineage cell pair numbers: $n$ = 1,378; 120; 848 for F; $n$ = 6; 6; 16 for G). Right: terminal cell types (colors) produced by the two daughter cells following development. Box plot: The center band is the median, box limits are the first and third quartiles, box length indicates IQR, and whiskers either 1.5 times the IQR or the minimum/maximum value if it falls within a factor of 1.5 times the IQR. Outliers not shown.

H    Correlation between chromatin activity transition and fate divergence following early cell divisions ($n$ = 90). Statistics: Pearson correlation.

I    Chromatin activity transition is associated with larger transcriptome divergence between two daughter cells. Bar plot shows the number of differentially expressed genes located within a 1-Mb interval centered on the GFP integration sites whose direction of expression change is consistent with that of chromatin activity between daughter cells following cell divisions exhibiting chromatin activity transition (red) or non-transition (blue). The mother cell names are shown on the *X*-axis. Statistics: Mann–Whitney $U$-test. The volcano plot in the inset shows the $\log_2$(fold change) (expression in the posterior cell divided by that in the anterior cell) and -$\log_{10}$($Q$-value) of all expressed genes following ABplaap cell division that produces an anterior daughter differentiating into mostly neuronal cells and a posterior daughter differentiating into exclusively skin cells. Arrows highlight two posterior-enriched genes (*lin-26* and *pax-3*) known to regulate skin differentiation.

J    Top: Schematic of ABalp-to-ABarp lineage fate transformation (purple arrow) induced by *lag-1(RNAi)* and the predicted changes in chromatin activity in cells from the ABalp lineage. Bottom: tree visualization of GFP expression integrated into a position in cells from the ABalp and ABarp lineages before and after lineage fate transformation. See Fig EV4A–C for the results of all seven integration strains.

K    Top: Schematic of MS-to-E lineage fate transformation (purple arrow) induced by *pop-1(RNAi)* and the predicted changes in chromatin activity in cells from the MS lineages. Bottom: tree visualization of GFP expression in cells from the MS and E lineages before and after lineage fate transformation. See Fig EV4D and E for the results of all five integration strains.

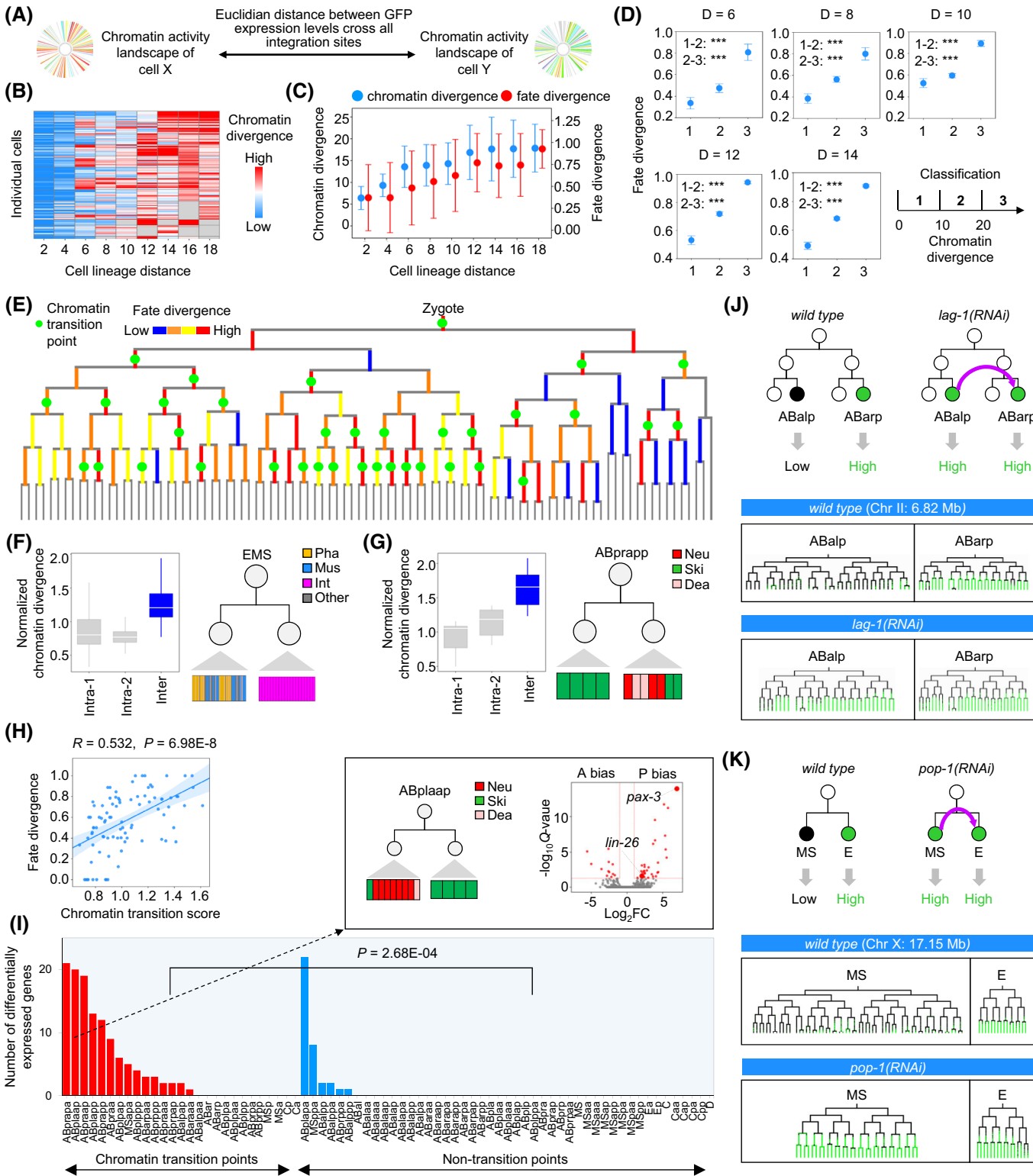

**Figure 3.**

distances. The results showed that a higher chromatin activity divergence was generally associated with a higher fate divergence, especially between cells at a modest lineage distance (from 6 to 14; Fig 3D). Thus, chromatin activity dynamics during lineage progression correlate with lineage-coupled cell differentiation.

An important process of lineage-coupled fate differentiation is anterior–posterior asymmetric cell division, in which a mother cell produces two daughter cells with distinct fates (Mizumoto & Sawa, 2007). Since most *C. elegans* cell divisions occur along the anterior–posterior body axis, this asymmetry is also termed

anterior–posterior fate asymmetry. We asked whether transitions in chromatin activity landscape following a cell division correlate with anterior–posterior fate asymmetry. A cell division is defined as a transition point for the chromatin activity landscape if the chromatin activity divergences between terminal cells generated by different daughter cells (inter-divergence) are significantly higher than those between terminal cells generated by the same daughter cell (intra-divergence; Fig EV3D). As exemplified by Fig EV3E, division of the ABplapp cell is identified as a chromatin activity transition point, since cells generated by its two daughter cells exhibit distinct chromatin activity landscapes. Of 90 early cell divisions, 36 (40%) showed significant chromatin activity transitions (Fig 3E and Dataset EV6). Within these transition points, we correctly captured the division of the EMS cell (Fig 3F), which is consistent with the knowledge that Wnt signaling induces an anterior–posterior fate asymmetry (Thorpe *et al*, 1997). We also identified transition points in many cases in which the cell division produces two daughter cells with distinct developmental fates. For example, a transition point was identified during asymmetric cell division of the ABprapp cell that produces an anterior daughter that exclusively differentiated into skin cells and a posterior daughter that differentiated into skin and neuronal cells and cells that undergo programmed cell death (Fig 3G). Globally, chromatin activity transition correlated significantly with the fate divergence between two daughter cells (Fig 3H), and 86% of the inferred chromatin activity transition points had concomitant high-fate divergence ($\geq 0.5$), which incidence was significantly higher than that observed in non-transition cases (Dataset EV6, $P = 2.95E-4$, Fisher's exact test).

Using scRNA-seq data (Packer *et al*, 2019), we directly examined whether transitions in chromatin activity landscape correlate with larger transcriptome divergences between the two daughter cells. Indeed, transcriptome divergences between daughter cells showing chromatin activity transitions were significantly larger than between those without transitions (Fig EV3F). Moreover, we explicitly tested whether genes near the position exhibiting differential chromatin activity were differentially expressed between two daughter cells in the expected direction, and the result supported the expectation (Fig 3I). Significantly larger numbers of differentially expressed genes were observed between two daughter cells following a cell division exhibiting chromatin transition than that of other cell divisions ($P = 2.68E-04$). For example, expression levels of two genes involved in skin differentiation (*pax-3* and *lin-26*) genes were significantly higher in the posterior daughter of ABplaap cell than the anterior daughter, concomitant with chromatin activity changes, in which the regions containing the two genes exhibited higher chromatin activity in the posterior daughter lineages (Labouesse *et al*, 1994; Labouesse *et al*, 1996; Thompson *et al*, 2016). Consistently, the posterior lineage differentiates exclusively into skin cells (Fig 3I, insert). Thus, the inferred chromatin activity transitions correlate with anterior–posterior asymmetry during lineage progression.

The outcome of lineage-dependent regulation is to specify the fates of progenitor cells according to their lineage identities. For example, the fate of ABalp is always to produce pharyngeal and neuronal cells, whereas the E progenitor cell invariantly differentiates into intestinal cells. First, we determined to what extent characteristic chromatin activity landscapes are established in cells from different lineages. We divided the entire cell lineage into smaller lineage groups and compared chromatin activity across all genomic positions between cells from different lineage groups. We found chromatin activity landscape differed in cells from different lineage groups (Fig EV3G), suggesting the establishment of lineage-specific landscapes. For example, quantification of the fraction of genomic positions at which the on/off state of chromatin activity was different between two cells showed that when dividing the cell lineage into 50 groups, chromatin activity at 37% of all positions (ranging between 7 to 77%) of cells from ABala lineages was different, on average, as compared to cells from all other lineages.

Next, we performed lineage fate perturbation experiments to test whether cellular chromatin activity changes accordingly once lineage fates are switched. We first used the ABalp and ABarp lineages to address this question because cells derived from the two lineages exhibited significantly different chromatin activity landscapes. RNA interference (RNAi) was used to knock down the function of the Notch effector *lag-1/CLS* (Moskowitz & Rothman, 1996), which induced an ABalp-to-ABarp lineage fate transformation (Fig 3 J). In *lag-1(RNAi)* embryos, the tree topology and characteristic programmed cell death of the ABalp lineage were distinct from normal ABalp but resembled those of the ABarp lineage (Fig EV4A and B), which confirmed the induction of lineage fate transformation. Concomitantly, in all seven integrants that were examined, GFP, which is usually silenced or weakly expressed in ABalp lineage-derived cells, was highly expressed, similar to the normal ABarp lineage (Figs 3J and EV4C). These results suggest that chromatin activity landscape and cell lineage fate are coupled. This coupling was confirmed in another developmental context. The fate of the MS lineage was switched to that of E by knocking down the *pop-1* gene (Lin *et al*, 1995), a Wnt signaling component (Figs 3K and EV4D). Concomitantly, in all five tested integrants, GFP expression was up-regulated in MS lineage-derived cells, resembling that of typical E lineage-derived cells (Figs 3K and EV4E).

Collectively, the above lineage-centric analyses support that chromatin activity landscape diversifies considerably across cells during early lineage progression and is systematically associated with lineage-coupled fate differentiation, including the global lineage-dependent diversification of cell fates, anterior–posterior fate asymmetry, and the establishment of lineage-specific fates.

## Tissue-based convergence of cellular chromatin activity landscape upon differentiation

The lineage-based mechanism archives a global patterning of cell fates in groups of lineage-related cells. Since most cell types are not monoclonal, this mechanism alone is not sufficient to generate the ultimate body plan. Complementarily, there is a tissue-based mechanism of cell differentiation in which cells from different lineages differentiate into the same tissue through the co-specification of identical fates in tissue precursors (Labouesse & Mango, 1999). This tissue-based mechanism is involved throughout embryogenesis because most somatic tissues, except the intestine, are derived from multiple cell lineages (Fig 4A).

As shown above, the chromatin activity landscape generally diversified across cells from different lineages, which raises the question of whether the landscapes of cells from distinct lineages converge according to tissue types. Two recent studies have shown that cellular gene expression converges upon tissue differentiation (Packer *et al*, 2019; Ma *et al*, 2020), raising the question of whether

chromatin convergence underpins the convergences of tissue fates and cellular transcriptomes. If it does, then chromatin activity divergences would be lower between cells of the same tissue (intra-tissue) as compared to between different tissues (inter-tissue). We classified all cells into five major tissue/organ types and found that the divergences were significantly lower for intra-tissue than inter-tissue comparisons (Fig 4B and Dataset EV7). For example, chromatin activity landscapes of skin cells from the Caa and Cpa lineages were highly similar, despite originating from distinct

lineages; the same was true for landscapes of body muscle cells (Fig 4C).

The fact that chromatin activity landscape diversifies across cells in different lineages and that similar landscapes are observed for tissue cells derived from distinct lineages indicates a tissue-dependent convergence during cell differentiation. Indeed, the chromatin activity divergences between differentiated intra-tissue cells were significantly lower than that between their mother cells (Fig 4D), indicating a progressive convergence of the landscape toward

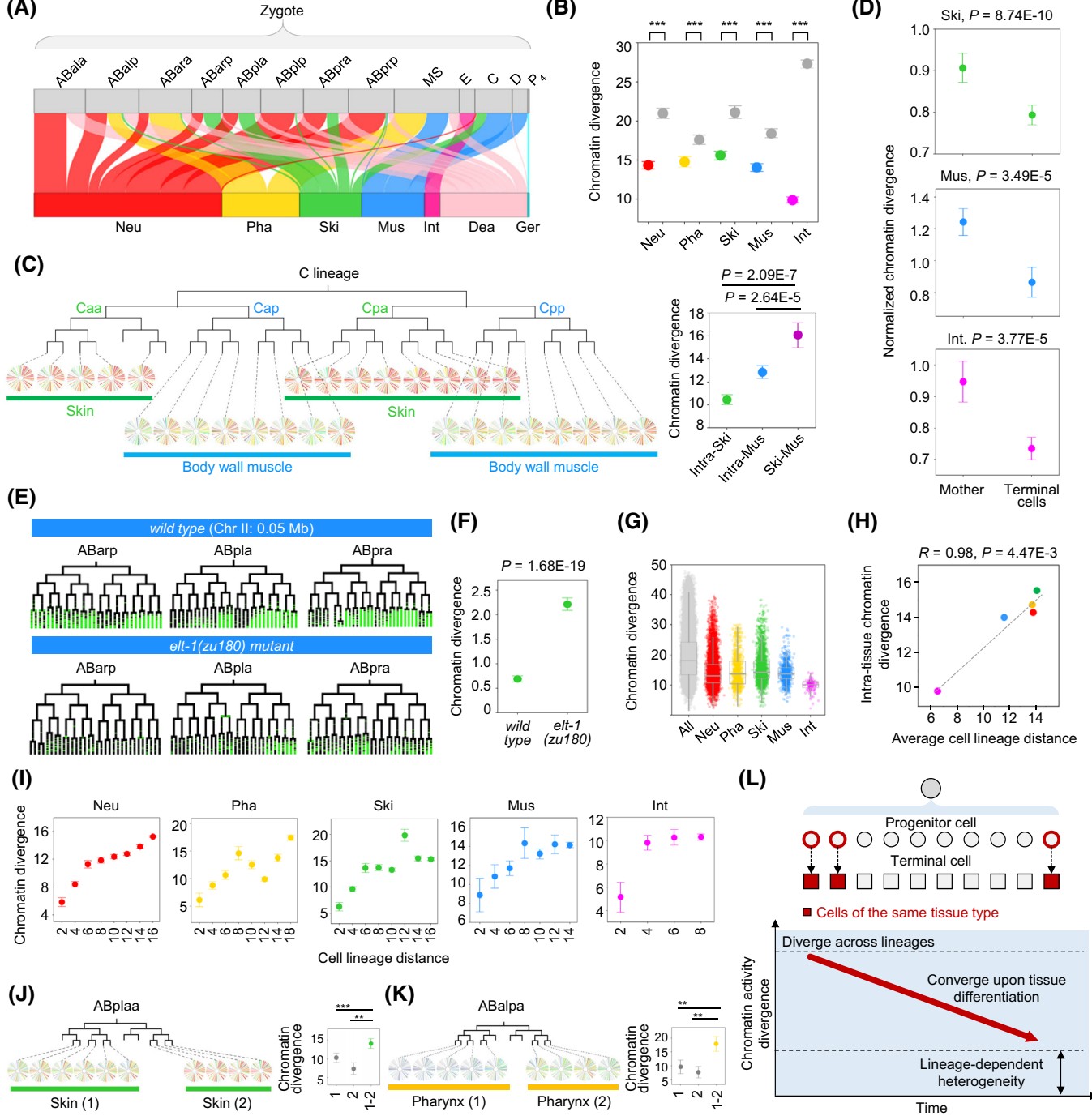

**Figure 4.**

**Figure 4.  Convergence and lineage-dependent heterogeneity of chromatin activity landscapes during tissue differentiation.**

A    The relationship between cell lineage identities (13 founder cells) and tissue fates. Dea, cell death; Ger, germ cell.

B    Comparison of chromatin activity divergences (mean ± 95% CI) between intra- and inter-tissue cells (gray) for each tissue (cell numbers, $n = 128$ for Neu, 44 for Pha, 75 for Ski, 44 for Mus, and 16 for Int). Statistics: Mann–Whitney *U*-test, ***$P < 0.001$.

C    Convergence of chromatin activity landscapes in skin and body wall muscle cells from the C lineage. Left: chromatin activity landscapes (Circos plot) of individual skin and muscle cells. Right: comparison of chromatin activity divergences (mean ± 95% CI) between skin cells (intra-skin, $n = 13$), between body wall muscles (intra-mus, $n = 16$), and between skin and muscle cells (ski-mus, $n = 29$). Statistics: Mann–Whitney *U*-test.

D    Compassion of chromatin activity divergences (mean ± 95% CI) between differentiated intra-tissue cells and between their mother cells. The neuronal and pharyngeal cells are not included because of a small number of differentiated cells at the 350-cell stage. Statistics: Mann–Whitney *U*-test.

E    Tree visualization of GFP expression at an integration site in cells from the ABarp, ABpla, and ABpra lineages before and after perturbing skin differentiation using an *elt-1* mutant. See Appendix Fig S3 for the results of all three integration strains.

F    Comparison of chromatin activity divergences (mean ± 95% CI) between ABarp-, ABpla-, and ABpra-derived skin cells ($n = 54$) before and after perturbing skin differentiation. Statistics: Mann–Whitney *U*-test.

G    Comparison of chromatin activity divergences between all cells (cell pair number: $n = 66,066$) and intra-tissue cells (cell pair numbers: $n = 8,128$ for Neu; 946 for Pha; 2,775 for Ski; 946 for Mus; 120 for Int). The center band is the median, box limits are the first and third quartiles, box length indicates IQR, and whiskers either 1.5 times the IQR or the minimum/maximum value if it falls within a factor of 1.5 times of the IQR. Each dot represents one cell pair.

H    Correlation between average lineage distance and chromatin activity divergence between intra-tissue cells for all tissues (different colors). Statistics: Pearson correlation.

I    Changes in chromatin activity divergences (mean ± 95% CI) following the increase in cell lineage distance between intra-tissue cells (cell pair numbers, from left to right: $n = 60$; 106; 193; 333; 609; 1,198; 1,990; 4,556 for Neu; $n = 25$; 46; 77; 133; 141; 110; 276; 735 for Pha; $n = 28$; 52; 72; 120; 286; 54; 303; 816 for Pha; $n = 18$; 33; 54; 28; 73; 48; 96 for Mus; $n = 8$; 16; 32; 64 for Int).

J, K    Two examples show lineage-dependent heterogeneity in chromatin activity landscapes of skin (J) and pharyngeal cells (K) originated from different cell lineages. In each example, the left penal shows the chromatin activity landscapes of cells from corresponding lineages (marked by different numbers), and the right panel shows the comparison of chromatin activity divergences (mean ± 95% CI) between cells from the same lineage group (1 and 2; cell numbers: $n = 6$ and 4 for J; 5 and 4 for K) and from different lineage groups (1–2; cell numbers: $n = 10$ for J; 9 for K). Statistics: Mann–Whitney *U*-test. ***$P < 0.001$; **$P < 0.01$.

L    Convergence and lineage-dependent heterogeneity of cellular chromatin activity landscapes during tissue differentiation. Top: tissue differentiation of terminal cells (rectangles) from progenitor cells (circles) derived from different cell lineages. Progenitor and terminal cells that differentiated into the same tissue type are marked in red. Bottom: changes in chromatin activity divergences between progenitor and terminal cells that differentiate into the same tissue type.

terminal tissue differentiation. Tissue fate perturbation experiments further demonstrated that the tissue-dependent converge of chromatin activity landscapes relies on tissue fate. Specifically, a mutant of the *elt-1*/GATA1 gene, a specifier of skin fate (Page *et al*, 1997), was used to abolish skin differentiation. Using three integrants, we found that, when skin differentiation was perturbed, chromatin activity in cells from major skin lineages dramatically changed, and the chromatin convergence was significantly affected as well (Fig 4E and F, and Appendix Fig S3). Together, these results reveal a chromatin basis for the tissue-based convergence of regulatory states in cells originating from diverse lineages.

**Chromatin activity landscape exhibits "memory" of lineage origins that contributes to cell heterogeneity**

While chromatin activity landscape converges according to tissue type, the extent of this convergence is highly variable (Fig 4G). For example, the chromatin activity divergences between neuronal cells were significantly higher than that between intestinal cells. Furthermore, within the same tissue, the divergences between some cells were notably higher than that between other cells. Given that cells of a tissue derive from different lineages, we tested whether lineage origin accounts for this intra-tissue chromatin heterogeneity. Our recent finding using cellular protein expression of transcription factors has revealed that many tissue-specific TFs show lineage-restricted expression, contributing to a lineage-dependent intra-tissue heterogeneity in gene expression (Ma *et al*, 2020). Here, we further tested whether chromatin changes underlie this pattern. Interestingly, cell lineage composition was highly predictive of the chromatin heterogeneity in each tissue, with tissue composed of cells from diverse lineages exhibiting higher heterogeneity ($R = 0.98$; Fig 4H). Quantifying intra-tissue chromatin activity

divergence as a function of cell lineage distance further showed that intra-tissue cells derived from distant lineages tended to exhibit higher divergences (Fig 4I); for example, discernible chromatin heterogeneity was evident in skin cells and pharyngeal cells from different lineages (Fig 4J and K). However, not all of the 364 traced terminal cells had completed cell differentiation, raising the question of whether such effects were stably maintained in terminally differentiated cells. We repeated the above analysis focusing on only post-mitotic cells within the traced cells, which are likely to have completed terminal differentiation (Dataset EV7). A similar result was obtained (Fig EV5A), confirming that the lineage effects on intra-tissue chromatin heterogeneity were also present in mature cells.

This finding then led to the question of whether lineage-dependent chromatin heterogeneity results in heterogeneity in global gene expression in addition to transcription factors (Ma *et al*, 2020). Using scRNA-seq data, we found that the transcriptome divergences between intra-tissue cells increased with cell lineage distances not only at the 350-cell stage but also at the 600-cell stage, when most cells have almost completed embryonic differentiation (Fig EV5B and C). To further test whether the observed lineage-dependent heterogeneity persists after embryogenesis, we analyzed lineage-resolved expression data at the first larval stage (L1) animals (Liu *et al*, 2009) to examine lineage effects on gene expression, from which we obtained supportive results (Fig EV5D).

Since the chromatin activity landscape exhibits both lineage dependence and tissue dependence, we assessed whether the chromatin-tissue association persists after considering the influence of lineage. To control lineage distance, chromatin activity divergences were compared between intra-tissue cells and the lineage distance-matched inter-tissue cells; this analysis showed that chromatin activity divergences between intra-tissue cells were significantly

lower than the control cells for all tissue types (Appendix Fig S4A). Furthermore, the relative pair-wise chromatin activity divergences between cells were quantified by normalizing the divergence to the average divergence of all cells at the same lineage distance. Unsupervised clustering of cells using these lineage effects controlled chromatin activity divergence revealed seven broad clusters, each of which was significantly enriched for cells of certain tissue types (Appendix Fig S4B and C, Dataset EV7). These results confirm a tissue-based chromatin convergence, even though the precise extent of that convergence is affected by lineage origin.

While a specific tissue type was enriched in most cell clusters, both pharynx and body wall muscle fates were co-enriched in cells from cluster 3. Interestingly, most of the pharyngeal and muscle cells in this cluster are from the MS lineage (Dataset EV7), suggesting a lasting influence of lineage effects on tissue differentiation. Although muscle fate was enriched in cells from cluster 4, it explained only a minority (39%) of cell fates. Given that cells in this cluster exhibit very diverse lineage and fate compositions (Dataset EV7), the implications of a relatively similar chromatin activity landscape being shared by these seemingly unrelated cells remain to be determined.

Taken together, we reveal tissue- and lineage-dependent dynamics of chromatin activity during tissue differentiation in which the chromatin activity landscapes of cells from distinct lineages converge on tissue-specific patterns but retain stable memories of each cell's lineage history, contributing to cell heterogeneity within a tissue (Fig 4L). This lineage-dependent tissue heterogeneity raises the possibility that the lineage origin of cells may drive the functional diversification of cell types within a tissue through a chromatin-based mechanism.

### Predetermination of chromatin activity landscape during left–right symmetry establishment

With a bilateral body plan, many cells of a given tissue are organized in a pattern having left–right (L–R) morphological symmetry. L–R cells are usually organized as pairs of symmetric lineages, in which all cells located on the left are derived from one lineage while the corresponding cells on the right are derived from another (Fig 5 A). In addition to being the same cell type, the vast majority of the cells on the left side are indistinguishable from their right-side counterparts in terms of anatomy and function (Sulston *et al*, 1983).

Given the high functional similarity between L–R cells, we determined whether they exhibit higher chromatin similarity than non-L–R cells. Some of the traced terminal cells are L–R cells or progenitor cells in the corresponding L–R symmetric lineages (L–R progenitors; Dataset EV8). Interestingly, chromatin activity divergences between pairs of L–R cell/progenitors were lower than those observed in other intra-tissue comparisons having a matched lineage distance for 91% of the cases ($P = 2.42E-21$; Fig 5B and C, Dataset EV8). Nevertheless, the L–R cells *per se* do not fully account for the higher chromatin similarity observed between cells of the same tissue. Chromatin activity divergences between intra-tissue cells were significantly lower than that between inter-tissue cells after removing the L–R cells (Appendix Fig S5A). Furthermore, this pattern was also evident in the cellular gene expression data from L1 stage animals (Liu *et al*, 2009), in which gene expression divergences between L–R cells were significantly lower than those

between other cells having matched lineage distance within a tissue (Appendix Fig S5B).

Intriguingly, while cells in many L–R pairs have distinct lineage origins, the lineage-dependent differences in cellular chromatin activity diminished, in which the divergences between L–R cell pairs at distinct lineage distances were highly comparable (Fig 5D). As exemplified by Fig 5E, chromatin activity divergences between L–R cells are independent of cell lineage distance, as the divergences between L–R cells/progenitors separated by eight generations (lineage distance = 16) are highly comparable to those separated by a less number of generations (lineage distance = 12, 8, and 4). Moreover, such a pattern was also observed by comparing gene expression divergence in the L1 stage animals (Appendix Fig S5C). This finding suggests that chromatin activity dynamics during symmetry establishment are distinct from regulation during general tissue differentiation.

Two mechanisms are possible: the predetermination of analogous chromatin activity landscapes in early progenitors of L–R cells, or a more robust convergence of chromatin landscapes during symmetry establishment. Upon comparing chromatin activity landscapes at different stages, the results favored the predetermination model. Out of all traced cells, 48 L–R pairs completed embryonic mitosis, and so likely represented cases where the regulation of symmetry establishment had been completed. We found that chromatin activity divergences were highly comparable between these differentiated L–R cells and between their mother cells ($P = 0.15$; Fig 5F). We were unable to use P*eef-1A.1::*GFP to compare chromatin in many early L–R progenitors because it was not expressed during very early embryogenesis (Fig EV1H). We instead used the promoter of another ubiquitously expressed gene (*nhr-2*) (Zacharias *et al*, 2015) to generate 13 integrants that expressed GFP and exhibited position effects in early embryos (Appendix Fig S5D and E). With these strains, we validated that chromatin activity divergences between L–R cell/progenitor pairs were significantly lower than in lineage distance and fate-matched cells (Appendix Fig S5F). Furthermore, the divergences between L–R progenitor cells at earlier developmental stages were similarly low as between L–R cells at the 350-cell stage (Fig 5G and H, and Dataset EV9), supporting the predetermination model. Consistently, the cellular transcriptomes of paired progenitors of L–R cells were broadly indistinguishable; cell identity assignment based on scRNA-seq data showed that the transcriptomes of L–R progenitor cell pairs (91.5%) and of differentiated L–R cell pairs (92.7%) were highly indistinguishable (Packer *et al*, 2019).

Together, single-cell analysis of chromatin activity landscapes reveals a predetermination of cellular chromatin during the establishment of L–R symmetry, in which highly analogous chromatin activity landscapes are programmed in the early progenitors of prospective L–R symmetric cells and maintained during later development (Fig 5I).

### Chromatin activity co-dynamics inform the functional coordination of the genome

Having examined the cellular dynamics of chromatin activity, we sought to investigate its genomic organization. Specifically, we examined whether concordant changes in chromatin activity across cells predict the functional relevance of genome regions.

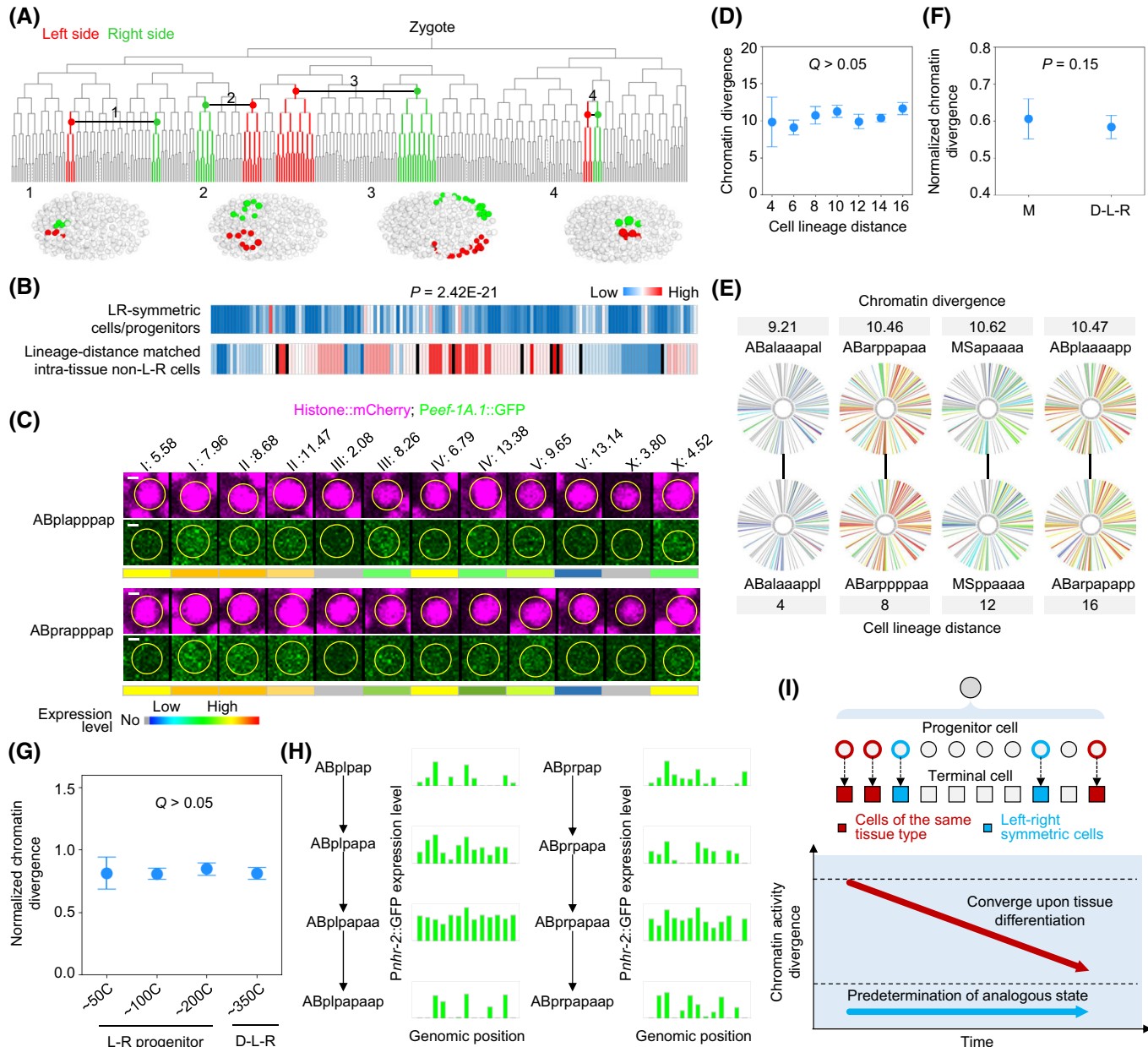

**Figure 5. Predetermination of chromatin activity landscape during L–R symmetry establishment.**

A  Developmental organization of L–R symmetric cells. Top: tree visualization of the cell lineage until the 350-cell stage. Horizontal lines link four pairs of L–R symmetric cell lineages (labeled as 1–4). Bottom: embryonic locations of corresponding cells.

B  Heatmap comparing chromatin activity divergences between cells in all pairs of L–R symmetric cells/progenitors and between control cells. Statistics: Wilcoxon signed-rank test, *n* = 149.

C  Micrographs compare the expression of ubiquitous mCherry (bottom) and GFP (top) integrated into 12 representative genomic positions between two L–R symmetric cells. Scale bar = 1 µm.

D  Comparison of chromatin activity divergences (mean ± 95% CI) between L–R symmetric cells with different cell lineage distances (cell pair numbers from left to right: *n* = 3; 16; 15; 18; 17; 62; 16). Statistics: Pair-wise Tukey-HSD *post hoc* test, Benjamini–Hochberg-adjusted *P*-value.

E  Figure shows chromatin activity landscapes, chromatin activity divergence scores, and cell lineage distances for four representative L–R symmetric cell pairs.

F  Comparison of chromatin activity divergences (mean ± 95% CI) between differentiated L–R symmetric cells (D–L–R, *n* = 48) and between their mother cells (M). Statistics: Mann–Whitney *U*-test.

G  Comparison of chromatin activity divergences (calculated using P*nhr-2*::GFP expression, mean ± 95% CI) between differentiated L–R cells at the 350-cell stage and between L–R progenitor cells at different developmental stages based on the approximate number of cells in the embryo (cell pair numbers: *n* = 19 for ~ 50C; 45 for ~ 100C; 86 for ~ 200C; 159 for ~ 350C). Statistics: Pair-wise Tukey-HSD *post hoc* test, Benjamini–Hochberg-adjusted *P*-value.

H  Bar plots show chromatin activity landscapes of cells in two cell tracks that lead to a pair of L–R symmetric cells. Arrow indicates the mother-daughter relationship.

I  Predetermination of chromatin activity landscape during L–R symmetry establishment. Figure organization is the same as Fig 4L. L–R symmetric cells/progenitors are marked in light blue.

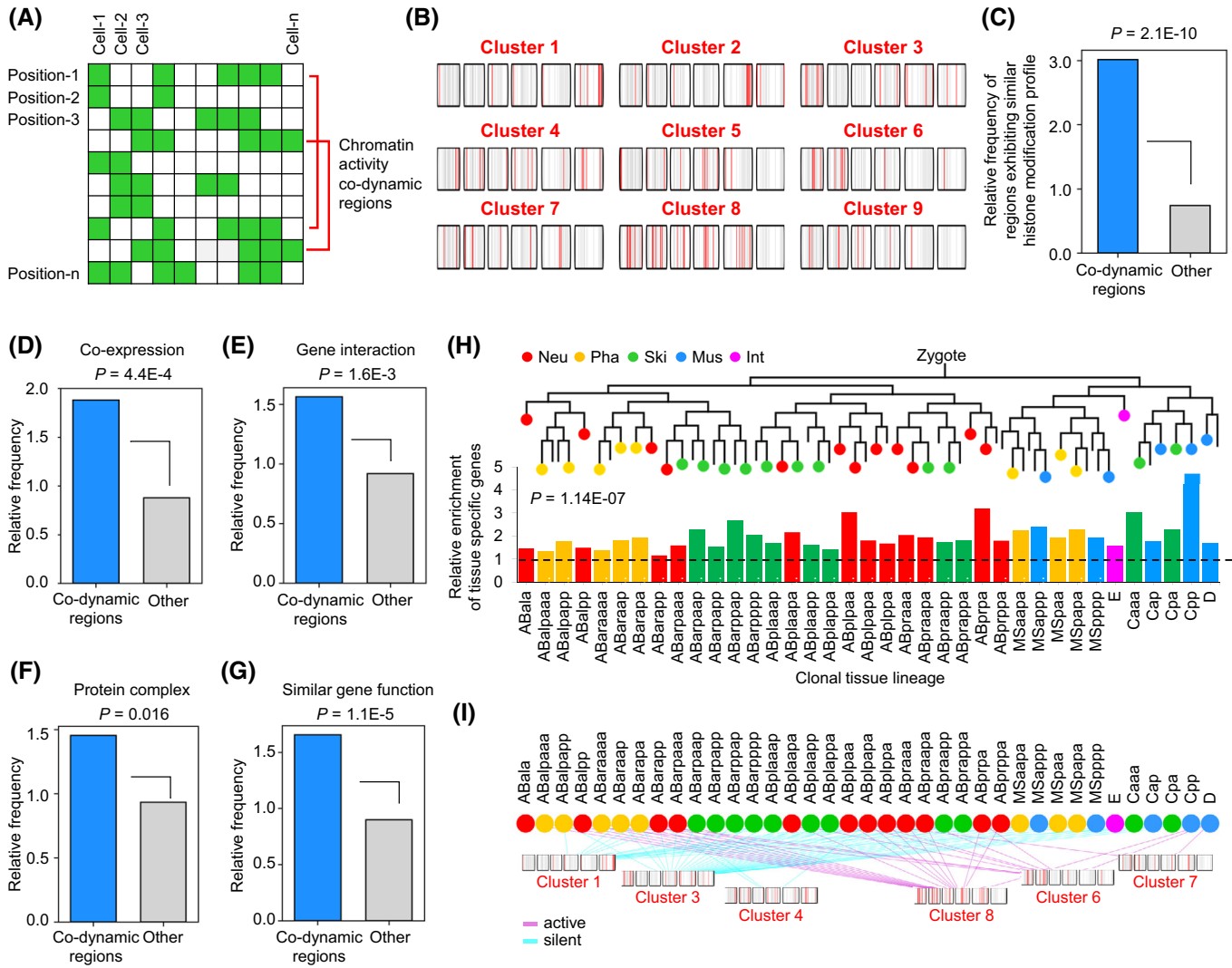

**Figure 6. Chromatin activity co-dynamic regions are enriched for functionally related genes.**

A    Strategy to identify chromatin activity co-dynamic regions. GFP expression pattern across cells is compared between integration sites to identify genomic regions that exhibit similar chromatin activity across cells (chromatin activity co-dynamic regions, connected by red lines).

B    Genomic locations of nine clusters of chromatin activity co-dynamic regions. Each block of the barcode represents a chromosome with vertical lines indicating the location of the reporter integration site. Chromatin activity co-dynamic regions within each cluster are marked in red.

C    Comparison of the relative frequency of pair-wise regions exhibiting similar histone modification profiles between chromatin activity co-dynamic regions and between non-co-dynamic regions. Two regions were considered to exhibit similar histone modification profiles if the Pearson correlation coefficient of the relative levels of 19 types of histone modifications is > 0.8.

D–G   Relative frequency of co-expressed genes (D), interacted genes (E), genes present in the same protein complex (F), and genes with similar functions (G) in all genes located with 100-kb centered on the reporter integration sites in the chromatin activity co-dynamic (blue) and non-co-dynamic regions (gray). Statistics: Hypergeometric test.

H    Top: tree visualization of clonal tissue lineages (colored circles, coded according to tissue type, $n = 37$). Bottom: relative enrichment of tissue-specific genes in genomic regions whose chromatin is activated as compared to those that are silenced in cells from each clonal tissue lineage. Names of the progenitor cells of clonal tissue lineages are shown on the $X$-axis. The dashed line indicates the expected enrichment score. Statistics: Wilcoxon signed-rank test.

I    Clusters of chromatin activity co-dynamic regions (barcodes) are enriched in all genomic regions that exhibit active (purple lines) or silent (cyan lines) chromatin activity in individual cells from corresponding clonal tissue lineages.

We determined to what extent different genomic regions exhibit similar chromatin activity changes by calculating the divergences across cells among the 113 genomic positions (Fig 6A) and identified nine clusters of genomic regions (720 pairs) exhibiting similar chromatin activity changes across cells, which we termed chromatin activity co-dynamic regions (Fig 6B and Dataset EV10). Interestingly, these activity co-dynamic regions exhibited similar histone modification profiles (Fig 6C), suggesting they are co-regulated. A small fraction (4%) of chromatin activity co-dynamic regions were linked on the chromosome at a distance of less than

one Mb; most (80%) co-dynamic regions were located on different chromosomes.

Do chromatin activity co-dynamics indicate functional relevance? We tested this possibility by examining whether genes near the regions tend to be functionally related. Three types of functional relevance between genes were tested as follows: co-expression, gene interaction, and functional similarity, and all of the results supported the hypothesis (Fig 6D–G and Dataset EV10). First, pairs of chromatin activity co-dynamic regions were significantly enriched for genes that are co-expressed across more than 900 conditions (Fig 6D). Second, genes located near pairs of the activity co-dynamic regions tended to interact with each other (Fig 6E) and were enriched for proteins that are present in the same complexes (Fig 6F). Finally, genes near pairs of activity co-dynamic regions tended to encode proteins with identical functional annotations (Fig 6G). Since chromatin activity co-dynamic regions are likely to be co-regulated (e.g., by histone modifications; Fig 6C), these findings raise the possibility that chromatin regulation drives the functional coordination of the genome.

Lastly, we tested whether chromatin activity co-dynamic regions coordinate the expression of regulatory genes during cell lineage differentiation. Since the chromatin activity landscape was associated with both lineage and tissue, we divided cells into 37 tissue lineages, each corresponding to a clonal lineage that differentiates mostly (> 85%) into a single tissue type (Fig 6H, top). We found that active chromatin regions in cells from each clonal tissue lineage were significantly enriched for genes that were preferentially expressed in the corresponding tissue as compared to the silent chromatin regions (Fig 6H, bottom). Moreover, the activation or silencing of specific clusters of chromatin activity co-dynamic regions was widely observed in cells from clonal tissue lineages (Fig 6I). These results suggest that co-regulation of chromatin activity invokes functionally related genes during lineage differentiation.

# Discussion

Many sequencing-based approaches efficiently dissect different aspects of the state of chromatin, such as DNA and histone modifications (Mikkelsen et al, 2007; Xie et al, 2013; Zhu et al, 2018), binding of regulatory proteins (Gerstein et al, 2010), accessibility (Cusanovich et al, 2018a), and spatial organization (Lieberman-Aiden et al, 2009). However, it is challenging to use these approaches to elucidate the functional state of chromatin in explicit single cells during in vivo development. First, the contribution of chromatin's biochemical or biophysical states to its activity is highly multifaceted. For example, the influence of histone modification on gene expression is highly context-dependent and often relies on other types of histone modifications (Wang et al, 2008; Karlic et al, 2010). Similarly, high chromatin accessibility does not always correspond to high activity (Arnold et al, 2013). This uncertainty complicates the interpretation of the regulatory role of chromatin if only a limited number of chromatin properties are evaluated. In this work, we quantified the position effects on a reporter gene that is highly responsive to chromatin environments as a direct functional measurement of chromatin activity. Since the same gene was introduced into many genomic positions and ultimate gene expression was used as a readout of the chromatin activity, the data described here represent the functional state of chromatin. Multiple pieces of evidence demonstrate that position effects on gene expression reliably indicate chromatin activity (Fig EV2). In addition, although the performance of some of the epigenomic methods mentioned above has been improved for analyzing low cell numbers or the chromatin state in single cells (Stevens et al, 2017; Cusanovich et al, 2018a; Cusanovich et al, 2018c; Zhu et al, 2018; Ai et al, 2019), significant challenges remain with assigning lineage identities to the measured chromatin states in individual cells. Here, we applied live imaging and direct lineage tracing to determine position effects on reporter expression in precisely traced and lineaged single cells, allowing us to infer the chromatin activity landscape in specific cells within an intact embryo (Fig 1). The functional nature, cellular resolution, and high cell coverage of the chromatin activity landscape provide a unique opportunity to systematically explore the implications of chromatin activity dynamics during in vivo development.

While chromatin state provides rich information to regulate gene expression and to specify cellular regulatory states, to what extent the regulatory processes that govern in vivo development are encoded in the chromatin of developing cells remains an open question. Lineage-resolved chromatin activity landscape combined with the extensive prior knowledge of single-cell biology of C. elegans development allows us to investigate the dynamics of chromatin activity during cell lineage differentiation in single cells. Through a multidimensional analysis of the lineage-resolved chromatin activity landscape, we found that chromatin activity dynamics correlate with lineage commitment, anterior–posterior asymmetry, tissue fate specification, cell heterogeneity, and bilateral symmetry establishment (Figs 2–5). This suggests that regulatory events in cell differentiation could be inferred from cellular chromatin. A previous study that profiled genome-wide maps of DNase I-hypersensitive sites in diverse human embryonic stem cells and adult primary cells revealed that the chromatin landscape reflects cell lineage relationships, cell fates, and cellular maturity (Stergachis et al, 2013). Thus, cellular chromatin provides highly specific information regarding fate patterning, paralleling the genetic programs of that process (Liu et al, 2009; Murray et al, 2012; Araya et al, 2014; Packer et al, 2019).

We provide three insights into chromatin activity dynamics during developmental and genome regulation. First, while the chromatin activity landscape of cells from different lineages converge according to tissue type, cellular chromatin encodes a "memory" of developmental history that contributes to heterogeneity within functionally related cells (Fig 4). It suggests that differences in developmental histories of cells contribute to cellular heterogeneity within a tissue type and, more generally, implies that the functional diversification of cells at the sub-tissue/cell type level is developmentally encoded in the cell lineage. A recent mouse study showed that cells belonging to a given tissue type possess characteristic chromatin accessibility patterns according to their location in the body (Cusanovich et al, 2018b). While the lineage identities of these cells remain to be determined, it is likely they originate from discrete cell lineages. Furthermore, in the mammalian nervous system, specific lineage-related neurons are reported to have higher functional relation in terms of microcircuit assembly and stimulus feature selectivity (Li et al, 2012; Yu et al, 2012). Thus, lineage-dependent cell heterogeneity appears to be evolutionarily conserved. The convergence of chromatin activity landscape during tissue differentiation

also implies that the pioneer transcription factors responsible for specifying tissue fate may play an essential role in shaping the characteristic landscapes and that the cooperative interactions between chromatin and pioneer factors may drive tissue differentiation (Zaret & Mango, 2016). Important follow-up studies include the following: identifying which pioneer factors are responsible for remodeling the chromatin activity landscape; how these pioneer factors co-specify chromatin activity in functionally related cells from discrete lineages; and determining how the lineage-dependent heterogeneity in chromatin activity is established.

Second, convergence and predetermination of the chromatin activity landscape are differentially utilized during general tissue differentiation and bilateral symmetry establishment (Figs 4 and 5). One possible advantage of the predetermination strategy is specifying primed analogous states in early progenitors of prospective L–R cells, even in cell pairs that are lineage-unrelated, allowing a robust generation of functionally equivalent cells on the different sides of the body. An intriguing question is when and how equivalent chromatin activity landscape is precisely specified in early progenitor cells that are not lineage-related. Since progenitor cells from different cell lineages generally exhibit divergent chromatin activity and gene expression programs (Packer *et al*, 2019; Ma *et al*, 2020), the specification of equivalent chromatin activity would be a regulated process. In rare cases, L–R symmetric cells exhibit functional asymmetry (Hobert, 2014). For example, a pair of morphologically and positionally symmetric neurons (ASEL/R) show differential chemosensory capacities (Pierce-Shimomura *et al*, 2001); this asymmetry is known to be primed in early progenitor cells through a chromatin-based mechanism (Cochella & Hobert, 2012). Such predetermined asymmetry was not evident in the chromatin activity landscapes of ASEL/R progenitor cells (ABalppppp/ABpraaappp), which exhibit a low chromatin activity divergence comparable to other L/R symmetric cells/progenitors from the same symmetric lineages (ABalppp/ABpraaa) that do not have known functional asymmetry ($P = 0.2$, $Z$-test, one-tailed). In addition to the asymmetric ASEL/R, the ABalppppp/ABpraaappp progenitors also produce two pairs of L–R neuronal cells, including the AUAL/R cells, which have been recently shown to exhibit molecular asymmetry (differential expression of *C32C4.16*) (Charest *et al*, 2020) and the ASJL/R cells, in which no functional asymmetry has been observed. It is thus possible that the regulation of L–R asymmetry may hinge on specific chromatin loci rather than the chromatin landscape as a whole. Indeed, differential chromatin decompaction of the *lsy-6* microRNA locus has been shown essential for priming ASEL/R asymmetry (Cochella & Hobert, 2012).

Finally, chromatin activity co-regulation might participate in coordinate functionally related genomic regions (Fig 6). Many functionally related genes, such as tissue-specific genes, tend to be clustered in discrete genomic regions (Lercher *et al*, 2002; Roy *et al*, 2002; Pauli *et al*, 2006); the co-activating/silencing of chromatin activity across the genome could thus provide an effective strategy for invoking related genes as a cohort during developmental regulation.

Due to low-throughput generation of randomly integrated transgenic animals, the genomic resolution of the present chromatin activity landscape is not high (0.88 Mb). In the future, this limitation could be improved by using CRISPR/Cas9-mediated genome editing to integrate reporter genes into a specific region of interest or throughout the genome at a higher resolution. In addition, the expression window of the *eef-1A.1* promoter in the integrated strains does not cover early embryonic cells, which prevented us from measuring cellular chromatin activity during very early embryogenesis. This limitation can be resolved by using other promoters that are responsive to chromatin environments and expressed ubiquitously during very early development. With suitable promoters, chromatin activity at any genomic position in any cell and at any developmental stage will be traceable using our single-cell approach.

# Materials and Methods

### *Caenorhabditis elegans* strains and culture

The genotypes of all *C. elegans* strains used in this study are listed in Dataset EV1. Some of the strains were obtained from the Caenorhabditis Genetics Center. Unless otherwise specified, all strains were cultured in incubators at 21°C on nematode growth media plates seeded with OP50 bacteria.

### Selection and verification of reporter strains

From a previously generated collection of integration strains(Frokjaer-Jensen *et al*, 2014), a total of 116 reporter strains were selected based on the following criteria (https://wormbuilder.org/old/?page_id = 182): (i) driven by the *eef-1A.1* promoter (previously known as *eft-3*), (ii) use of GFP as the fluorophore, (iii) containing a nuclear localization signal, and (iv) with a unique integration site. Three strains (EG8880, EG8912, and EG8860) exhibited severe growth and developmental defects and were removed. Each of the 113 strains was crossed with the JIM113 strain that ubiquitously expressed mCherry in the nucleus for lineage tracing, which resulted in a collection of 113 dual-fluorescent reporter strains (Dataset EV1).

The integration sites of P*eef-1A.1*::GFP in all transgenic strains have been characterized previously. To further ensure the accuracy of the integration sites and copy numbers, 18 strains were randomly selected, and the integration sites were reexamined using inverse PCR following a previously established protocol (Frokjaer-Jensen *et al*, 2014). Genomic DNA was extracted and digested overnight with the restriction enzyme DpnII that cut at a unique site located in the MOS 1 vector and potential sites located near the integration site in the *C. elegans* genome. Next, the digested DNA was circularized by T4 ligase during a 2- to 4-h incubation. Two rounds of PCR were performed with the circularized DNA containing the flanking sequences as the template and with two pairs of nested primers targeting the MOS 1 sequence. PCR products were gel-purified and sequenced. The sequences were then aligned to the *C. elegans* genome (WBcel235/ce11) to identify the sequence of the regions flanking the integration site. For all 18 tested strains, integration sites were identical or within 1 kb of the previous annotation; additionally, only one integration site was identified in all examined strains (Fig EV1B).

### Embryo mounting and 3D time-lapse imaging of embryogenesis

A previously established procedure was used to prepare and mount early *C. elegans* embryos, with minor modifications (Bao & Murray, 2011). Briefly, six young adult worms with one row of eggs in the

gonad were picked and transferred onto a Multitest slide (MP Biomedicals) with a droplet of M9 buffer, and then, the worms were cut open to release early embryos. Two- to four-cell stage embryos were identified under a dissecting microscope (Nikon SMZ745) that were then transferred into a droplet of egg buffer (~ 2 µl) containing 20-µm polystyrene microspheres (PolyScience) on a coverslip (Fisherbrand) using an aspirator tube assembly (Sigma-Aldrich). The various positions of embryos were adjusted to be arranged in a few clusters, with each cluster containing two to three embryos (fit with one imaging field). Finally, an $18 \times 18$ mm coverslip was placed on top of the droplet and the slide was sealed with melted Vaseline.

3D time-lapse imaging was performed using a spinning-disk confocal microscope (Revolution XD) with an inverted microscope body (Olympus IX73), a spinning-disk unit (Yokogawa CSU-X1,) XYZ stage with Piezo-Z positioning (ASI PZ-2150-XYZ), an integrated solid-state laser engine (Coherent; 50 mw at 488 nm and 50 mw at 561 nm), and an Electron Multiplying Charge-Coupled Device (EMCCD; Andor iXon Ultra 897). Images were taken at 20°C using the multidimensional acquisition module of MetaMorph software (Molecular Devices) under a 60× objective (PLAPON 60XO, N.A. = 1.42). Images were recorded for at least 240 time points at a temporal resolution of 75 s, and at each time point, three slide positions with two to three embryos were scanned for 30 $Z$ focal planes with 1 µm spacing. Laser power and exposure time for mCherry (561 nm) and GFP (488 nm) were optimized to minimize photodamage while maintaining a high signal-to-noise ratio. Laser power for both mCherry and GFP was increased 3% for every $Z$ plane when the focal plane went deeper into the sample to partially compensate for the decay of the fluorescence signal over $Z$ focal panes. The laser power and exposure time used for mCherry and GFP for the first $Z$ plane were 8% for 50 ms and 8% for 20 ms, respectively. All wild-type embryos ($n > 50$) imaged with this parameter hatched at a time that was comparable to embryos without laser excitation without obvious phenotypic abnormalities. For individual time points, the two-channel image series were organized as 3D tiff stack images and were directly used for cell identification, tracing, lineage construction, and quantification of single-cell reporter expression.

## Cell identification, lineage tracing, and manual curation

Image series were processed with StarryNite software (Santella *et al,* 2010; Santella *et al,* 2014) to reconstruct *de novo* the embryonic cell lineage by automated cell identification and tracking. Automated cell identification was performed using a hybrid blob-detection algorithm to segment individual nuclei in the 3D image stacks based on the ubiquitously expressed mCherry fluorescence signal that localizes to the nucleus (Santella *et al,* 2010). Next, automated cell tracing was performed using a semi-local neighborhood-based framework to link all cells at a preceding time point to those at the subsequent time point, and if cell division occurs, a mother cell is linked to two daughter cells (Santella *et al,* 2014).

Raw cell identification and tracing results were systematically inspected and curated manually to ensure high accuracy using the AceTree software (Katzman *et al,* 2018). While the accuracy of automated cell detection and tracing by StarryNite software is high (> 99%), the accumulative nature of the errors affects the accuracy

of cell lineage results (Santella *et al,* 2014). Hence, a systematic correction of lineage errors, especially those that occur in the early developmental stage, is indispensable. AceTree software provides an interface for visualizing the traced cell lineage as a binary tree structure and links all cells on the tree to the raw 3D images. With this function, users can identify potential lineaging errors in the tree, inspect the cell relationships on the raw images, and finally modify the identification or tracing results when necessary. Because the *C. elegans* cell lineage is invariant, the vast majority of errors can be efficiently captured by visual or computational screening of unusual lineage topologies. Detailed procedures for error detection and correction were described previously (Du *et al,* 2015).

The lineage identities of all traced cells were determined and assigned a unique name according to Sulston's nomenclature (Sulston *et al,* 1983). Cell identities were first determined for all cells (ABa, ABp, EMS, and $P_2$) in the 4-cell stage embryos based on the stereotypical arrangement of cells in the embryo and the timing of cell divisions. Specifically, ABa and $P_2$ are located in the anterior and posterior parts of the embryo, respectively, and ABa and ABp divide earlier than EMS and $P_2$ cells. Then, the names of their descendants were determined according to the mother cell name and cell division pattern. All cell divisions fall into three broad categories: anterior–posterior (a/p), left–right (l/r), and dorsal–ventral (d/v), according to the orientation of cell division relative to the body axis. In general, the full name of a mother cell is propagated to the daughter cells with an additional letter specifying the cell position of the daughter cell relative to the body axis following cell division of the mother. For example, ABal specifies the daughter cell of ABa that is located on the left side following the l/r division of the ABa cell, and ABala specifies the daughter cell of ABal that is located anteriorly following the a/p division of the ABal cell. Except for the few early progenitor cells (MS, E, C, D, $P_3$, $P_4$, Z2, and Z3) to which a particular name was assigned to highlight their developmental properties, the name assignment of all cells followed this general rule. Detailed nomenclature information is described elsewhere (Sulston *et al,* 1983; Santella *et al,* 2010). Using the aforementioned image bioinformatics, the embryogenesis was digitized at cellular resolution and the cell lineage identities of all traced cells were determined.

## Quantification of reporter expression in lineaged cells

Each strain that was used to quantify the positional effects of reporter expression carried two nucleus-localized fluorescent proteins. The ubiquitous mCherry was used for the aforementioned cell lineage reconstruction, and the fluorescent intensity of GFP integrated into a specific genomic position was used to measure the reporter expression level in each traced nucleus simultaneously. The segmentation of each nucleus at each time point and tracing of nuclei across time during the lineage construction step facilitated a direct quantification of GFP expression in each lineage-resolved cell.

### Quantification of raw GFP expression

Raw GFP expression was calculated as the average intensity of all pixels within each identified nucleus at each time point minus the average intensity of the local background. The average pixel intensity in the center $Z$ plane of each nucleus was used to approximate the expression level in the nucleus. The background signal for each

nucleus was estimated using a previously described method in which the average pixel intensity was calculated within an annular area between 1.2- and 2-radius from the centroid of the nucleus. Nearby nuclei that overlapped with the annular area were not included in the background measurement (Murray *et al*, 2008). GFP intensities of the same cell at multiple time points were averaged, representing cellular average GFP expression abundance. Because the nucleus morphology at the time points immediately before and after cell division is not spherical (which would affect the accuracy of intensity measurements), the values at these time points were excluded when calculating the average GFP expression in a cell.

### Compensation for depth-dependent attenuation of fluorescence intensity

GFP levels were adjusted by compensating for the depth-dependent attenuation of fluorescence intensity. A fundamental problem associated with 3D fluorescence confocal imaging is the attenuation of light with depth, caused by the absorption and scattering of both the excitation and fluorescence light (Kervrann *et al*, 2004). Consequently, without adjustment, the measured fluorescence intensities of cells reside deeper in the embryo (far from the microscope objective) and are significantly weaker than those located in shallower slices. This effect could significantly obscure a reliable comparison of GFP expression between single cells in an embryo, since the equivalent cells are located at different depths (Z planes) relative to the objective during imaging (Fig EV1C). Although the attenuation effect was partially compensated for during imaging by increasing the laser power of the excitation light with depth, this effect was still present in the acquired images. This phenomenon is best illustrated by comparing GFP expression in equivalent cells between embryos of the same strain (experimental replicates) that are oriented differently during imaging. There are two types of embryo orientation at the 350-cell stage in which either the ventral or the dorsal side is placed near the objective (termed VNO and DNO, respectively). As shown in Fig EV1D, cellular GFP intensity was highly consistent between embryos with identical orientation (average Pearson correlation coefficient $R = 0.85$). However, the intensity differs considerably between embryos with different orientations (average Pearson correlation coefficient $R = 0.22$), especially for those cells located far from the center Z plane of the embryo. This discrepancy allowed us to model and correct the residual attenuation effect. Using the information of each cell's Z position and GFP intensity, we applied various attenuation factors per Z plane ($\alpha$) to adjust the GFP intensity at any Z plane to the center plane ($Z = 15$) using the equation $I_i = I_c \cdot (1 + \alpha)^{(i - c)}$, where $I_i$ and $I_c$ specify the GFP intensity of the cell at plane $i$ and the center plane, respectively. The performance of each $\alpha$ was evaluated by quantifying the correlation coefficient of cellular GFP expression between experiment replicates with different embryonic orientations. We used 49 embryos of 13 transgenic strains to model the performance of adjustment and determined whether the attenuation effect depended on the magnitude of fluorescent intensity. We found that the attenuation was significant when the cellular GFP intensity was greater than seven and that $\alpha = 0.054$ yielded the best performance regarding adjustment of the attenuation (Fig EV1E). These parameters were applied to all cells in all embryos, which dramatically removed the residual attenuation effects (see Fig EV1F for representative examples). This adjustment of GFP intensity facilitates a reliable inter-cell comparison of GFP expression in an embryo, especially when the cells are located in significantly different Z planes.

### Quantification of instantaneous GFP expression

Due to the high stability of GFP protein (with a half-life of over 20 h in mammalian cells) and a fast cell cycle progression during embryogenesis (median cell cycle length = 42 min until the 350-cell stage), the measured GFP intensity in a cell consisted of both GFP inherited from the previous cell cycle and GFP expressed in the present cell. Since GFP expression was continuously imaged at a high temporal resolution, the inherited and newly expressed GFP was distinguished by subtracting the GFP intensity of mother cells from the corresponding daughter cells. Specifically, GFP intensity value at the time point before the cell division of the mother cells was used to represent the GFP expressed in the mother cell and was subtracted from the value of the daughter cells, which assumed that the reporter is not expressed during cell division. We used strains with the mCherry lineaging marker but not the GFP transgene as the control and quantified cellular GFP intensity in 20 embryos to model the distribution of GFP intensity in non-expression cells. A cut-off of 6.36 ($Q < 0.01$) at which the false discovery rate was 6.9e-5 for cells in control embryos was used to refine GFP expression levels. For cells with an intensity lower than the cut-off, the expression level was set to zero; otherwise, the cut-off value was subtracted from the cellular GFP intensity to represent instantaneous expression level in a cell, which was then $\log_2(X + 1)$ transformed. Instantaneous GFP expression levels were used to represent chromatin activity.

## Construction of single-cell chromatin activity landscape

Chromatin activity landscape of individual cells was constructed by integrating GFP expression levels across all genomic positions in cells with the same lineage identity (equivalent cells). Because GFP expression at each integration site was quantified for multiple embryos (ranging from 2 to 8), expression levels were integrated across experimental replicates. In this integration, only cases in which GFP was expressed (with a value > 0) in more than 60% of replicates were considered to be expressed, and the instantaneous expression levels (untransformed) were averaged, and $\log_2(X + 1)$ transformed to represent consensus chromatin activity at a genomic location in a cell. Otherwise, the chromatin activity was set to zero.

## Comparison of position effects on GFP expression with chromatin features

### Definition of arm and center regions

Chromosome regions located within 20% of a chromosome at each end were defined as arm regions; the rest of the regions were defined as center regions.

### Histone modification

We predicted GFP expression levels at each integration site using the combination of 19 types of histone modifications (Ho *et al*, 2014). Histone modification datasets were downloaded from the ModENCODE project Web site, and only histone modifications of embryonic datasets ($n = 56$) were used to ensure a comparison of histone modification and GFP expression at the comparable developmental stage. Each dataset was normalized by calculating the Z

score $Z = (x − μ)/α$ for each region across the genome, where μ denotes the mean and α denotes the standard deviation of the modification levels. The level of each histone modification at the transgene integration site was determined and was used to represent the modification level of each P*eef-1-A.1*::GFP cassette by averaging the modification level of all genomic regions that overlapped with a 1-kb region centered on the integration site of the expression cassette. GFP expression levels were predicted by a linear model that considers all types of modifications. If multiple datasets are available for the same histone modification, the dataset that yielded the best performance was used.

### Chromatin accessibility

Accessibility chromatin regions were defined based on a previous study in which ATAC-seq was applied to identify accessible chromatin at several developmental stages (Janes *et al*, 2018). An integration site was defined as a location in accessible chromatin if an ATAC-seq peak overlapped with a 1-kb region centered on the integration site.

### Chromatin state/domain

Ho *et al* (2014) classified the *C. elegans* chromatin into 16 states, with states 1–3 corresponding to transcriptional activation and states 10–13 corresponding to transcriptional silencing. In addition, Evans *et al* (2016) segregated the chromatin into 20 states, in which states 1–5 were defined as chromatin domains with high activity and states 16–20 were defined as with low activity. A closer examination revealed that states 17, 18, and 20 were more robustly associated with gene silencing; we thus used these three states to represent silent chromatin. For both datasets, we compared the average expression levels of P*eef-1A.1*::GFP located in active states to those located in silent states to determine whether reporter expression is consistent with chromatin activity.

### LAD

LAD information was obtained from a previous study that determined regions associated with LEM-2, an inner nuclear membrane protein, in mixed-stage *C. elegans* embryos (Ikegami *et al*, 2010).

## Comparison of position effects on GFP expression with genetic features

### Definition of intragenic and intergenic regions

An integration site was defined as being located in the intragenic region if the annotated integration site is located within the gene body of any protein-coding gene; otherwise, the reporter was classified as being located in the intergenic region.

### Comparison of GFP expression with endogenous genes

A previously generated whole-embryo time-course transcriptome at high temporal resolution was used to represent average expression levels of endogenous genes (Hashimshony *et al,* 2015). Gene expression at stage 190 min past the first cell division that was comparable to the 350-cell stage was used for comparison. For each transgene integration site, the expression levels of all endogenous genes whose transcription start site is located in an interval centered on the transgene integration site were averaged to represent the expression potential of nearby genes. The transcript abundances were

$log_2(X + 1)$ transformed, and intervals of various sizes ranging from 5 kb to 2 Mb were analyzed.

### Comparison of GFP expression with endogenous genes in single cells

A previously generated single-cell transcriptome data of *C. elegans* embryogenesis were used to determine the correlation between the GFP expression and the expression of endogenous genes in equivalent cells (Packer *et al,* 2019). The expression potential of endogenous genes was quantified as the fraction of expressed genes (transcripts per million, TPM > 0) in a 500-kb genomic interval centered on the GFP integration sites. Because the lineage identity was not fully resolved for all cellular transcriptomes, only cells that had been assigned a unique identity ($n = 38$) or two possible identities ($n = 267$) were used for analysis.

## Perturbation of writers of histone modifications

The effects of H3K9me3 on chromatin activity were assessed using *set-25(n5021)*, a mutant of *set-25* (*G9a/EHMT2* homolog) that encodes the histone methyltransferase responsible for H3K9me3 methylation. A previous study has shown that H3K9 tri-methylation is abolished in *set-25(n5021)* (Towbin *et al,* 2012). The effects of H4K16ac on chromatin activity were assayed by performing RNAi against *mys-1* (*KAT5* homolog), a member of the MYST family histone acetyltransferase complex that adds acetylation to H4K16 (Lau *et al,* 2016). For each experiment, representative integration strains showing an enrichment of the corresponding histone modification in the 1-kb region centered on the integration sites were used to compare GFP expression levels before and after perturbation.

## Analysis of the dynamics and implications of the chromatin activity landscape during cell lineage differentiation

### Calculation of cell lineage distance

A previously described strategy was used to calculate the lineage distance between two cells as the number of cell divisions that separate these two cells from their lowest common ancestor (Du *et al,* 2015) (Fig EV3A).

### Definition and quantitative comparison of developmental fates between progenitor cells

The fate of a progenitor cell was defined retrospectively as the combinatorial pattern of tissue types it produced following development (Fig EV3B). The 671 embryonic terminal cells were first classified into seven tissue/organ categories: neuronal system (Neu; including neurons and glial cells), pharynx (Pha), skin (Ski), body wall muscle (Mus), intestine (Int), and germ cell (Ger). Cells that did not fall into any of these categories and that undergo programmed cell death were not considered in the analysis. The developmental fate of each progenitor cell was expressed as a pattern describing the tissue type of each terminal cell in the lineal order. The similarity was quantified by aligning the two patterns and comparing tissue fates between each lineal equivalent terminal cell (e.g., comparing Xapap to Yapap cells between progenitor cells X and Y). A score of 1 was assigned if two lineal equivalent cells belonged to the same tissue type; otherwise, 0 was assigned. The similarity scores were averaged across all terminal cells and were

used to represent overall fate similarity. In cases where two progenitor cells generated different numbers of terminal cells due to different rounds of cell divisions in intermediate cells, the number of cells and tissue fates in the smaller lineage was expanded accordingly in the corresponding branches to match the larger one. This strategy is illustrated in Fig EV3B.

### Quantification of chromatin landscape divergence

Chromatin activity divergences between cells were quantified as the Euclidean distance between chromatin activity (GFP expression levels) across all integration sites using the equation, $d(x,y) = \sqrt{\sum_{i=1}^{n}(x_i - y_i)^2}$ where $x$ and $y$ denote the vector of chromatin activity across 113 positions in the corresponding cells. For all chromatin activity divergence measurements, we used single cells as the basic unit to calculate the divergence, and the divergence values were averaged accordingly based on what cell groups were being compared.

### Identification of transition points of chromatin activity landscape

A retrospective approach was used to compare chromatin activity divergences between cells from two daughter cell lineages to determine whether the division of the mother cell leads to significant changes in chromatin activity landscape between two daughter cells. This assumption is based on the parsimony principle, which has been widely used in biology in which a minimal set of regulatory events are used to explain the observed outcomes. For each pair of daughter cell lineages generated by a progenitor cell, chromatin activity divergences were calculated between cells from each daughter cell lineage (intra-daughter–lineage chromatin activity divergence) and were compared with chromatin activity divergences between cells from different daughter cell lineages (inter-daughter–lineage chromatin activity divergence; Fig EV3D). A progenitor cell division was defined as a transition point of chromatin activity landscape if the inter-daughter–lineage divergences were significantly larger (Mann–Whitney $U$-test, Benjamini–Hochberg-adjusted $P < 0.05$) than any of the intra-daughter–lineage divergences. The chromatin activity transition score of a cell division was measured as the average value of the inter-divergence divided by each intra-divergence.

### Clustering of cells based on chromatin activity landscape

Pair-wise chromatin activity divergences were first calculated between 364 traced terminal cells, and the divergence between each cell pair of cells was then normalized to the average divergence of all cell pairs at the same lineage distance, which controlled for the influence of lineage distance between cells on chromatin activity divergence. Groups of cells with similar chromatin activity landscapes were identified by hierarchical clustering using the normalized cell–cell divergence matrix with the following parameters: Euclidean distance as the distance metric, average linkage clustering as the linkage selection method, and distance threshold = 4.5.

### Analysis of single-cell gene expression data

For single-cell transcriptome data (Packer et al, 2019), the Jenssen–Shannon distance was measured to quantify divergence, as was done in the original study. Unless otherwise noted, only cells assigned a unique or two possible lineage identities were used for analysis. Differentially expressed genes between two cells/

annotations were identified using the VisCello tool provided by the original study (github.com/qinzhu/VisCello.celegans) (Packer et al, 2019), and genes with a $\log_2$ fold change > 2 and a $Q$-value < 0.05 were defined as differential expression. If lineage identity is indistinguishable between two cells, the number of differentially expressed genes was defined as 0. For the identity-resolved single-cell gene expression dataset at the L1 stage (Liu et al, 2009), divergence was quantified by measuring the Euclidean distance using the binary expression matrix.

### L–R symmetric cells

The list of L–R symmetric cells and progenitor cells was defined by Sulston et al (1983).

## Lineage fate transformation and tissue fate perturbation experiments

Two pairs of cell lineages were selected, ABalp-ABarp and MS-E, and multiple reporter integration strains exhibiting differential GFP expression in descendant cells from corresponding lineages were used to examine the relationship between chromatin activity landscapes and lineage fates. Transformations of the ABalp lineage fate to that of normal ABarp and the MS lineage fate to that of normal E were induced by performing RNAi-mediated knockdown of lag-1/CLS and pop-1/TCF genes, respectively (Lin et al, 1995; Moskowitz & Rothman, 1996). RNAi treatments were performed on seven (lag-1 knockdown) and five (pop-1 knockdown) strains in which GFP was integrated into distinct genomic positions. Only embryos showing evidence of lineage fate transformation were used for comparing GFP expression between cells from corresponding lineages.

Perturbation of skin fate was performed by using a loss-of-function mutant of elt-1/GATA1, a skin fate specifier. As shown previously, Loss of elt-1 leads to extra neuronal (from the ABarp lineage), muscle (from Caa and Cpa lineages), and other cell types (from the ABpla lineage) at the expense of skin cells (Page et al, 1997). Furthermore, forced ectopic expression of elt-1 is sufficient to transform most embryonic cells into skin cells (Gilleard & McGhee, 2001). We used the elt-1(zu180) allele, which contains a stop codon within the first zinc finger domain, to induce fate changes in skin cells. To examine the influence of tissue fate perturbation on chromatin activity landscape, we compared changes in GFP expression in cells from three major skin lineages (ABarp, ABpla, and ABpra) using three integration strains in which GFP is expressed in cells from the aforementioned lineages.

### RNAi

RNAi experiments were performed using a standard feeding protocol (Kamath et al, 2001). All RNAi clones were from the C. elegans RNAi feeding library constructed by Julie Ahringer's group (Source BioScience). The insert of individual clones used in this study was verified by end sequencing. For RNAi treatment, 10 worms synchronized at the L1 stage (P0) were transferred to RNAi plates containing 3 mM IPTG and that had been seeded with the corresponding bacteria expressing the double-stranded RNA against the target gene. After several days of RNAi exposure, embryos of P0 or P1 animals were subjected to imaging and analysis.

## Analysis of the genomic organization of chromatin activity

### Identification of chromatin activity co-dynamic regions

Pair-wise divergences (Euclidean distance) of chromatin activity among cells were calculated between all reporter integration sites. The matrix was used to identify clusters of genomic positions by hierarchical clustering using the following parameters: Euclidean distance as the distance metric, average linkage clustering as the linkage selection method, and distance threshold = 65.

### Enrichment of functionally related genes

(i) Co-expressed genes were identified using a comprehensive dataset of global gene expression changes across diverse genetic and environmental conditions (15,204 genes across 979 conditions) (Stuart *et al*, 2003). Pair-wise Spearman rank correlation of gene expression across all conditions was calculated between all genes, and a pair of genes was defined as being co-expressed if the correlation coefficient ranked among the top one-third of all values (only gene pairs with a positive correlation coefficient were considered). (ii) Two sources of gene interaction data were used to identify interacted genes. First, a curated list of gene interactions (including genetic, physical, and regulatory interactions) was downloaded from WormBase (c_elegans.PRJNA13758.WS270.interactions.txt.gz). Second, a dataset of 612 putative *C. elegans* protein complexes generated by analyzing large-scale chromatographic fractionation-mass spectrometry (CF-MS) data was used to analyze the physical interactions between genes (Hu *et al*, 2019). (iii) Genes with similar functions were identified using the WormCat database, in which all *C. elegans* genes are classified into hundreds of non-redundant functional categories based on physiological functions, molecular function, phenotype, subcellular location, and other information (Holdorf *et al*, 2020). We used the Category 3 annotation (455 specific categories) and excluded four categories annotated as unknown for the analysis. Genes located within a 100-kb region centered on GFP integration sites were used to analyze the relative enrichment of functionally related genes.

### Identification of clonal tissue lineages

All embryonic terminal cells ($n = 671$) were classified into five major tissue types: the neuronal system, pharynx, skin, body wall muscle, and intestine. Then, the clonal analysis was performed to identify progenitor cell lineages that differentiate mostly into a given tissue type. To do so, we performed a bottom-up analysis to quantify the probability of each progenitor cell differentiating into a specific tissue type. For each progenitor cell, this probability was quantified recursively by averaging the score of its two daughter cells. Finally, the highest scoring clone with a probability > 0.85 that produces $\geq 3$ terminal cells belonging to the same tissue type was defined as a clonal tissue lineage.

### Tissue-specific genes

Lists of embryonic tissue-specific genes were obtained from a recent study that sequenced the transcriptome of major *C. elegans* tissues and organs based on fluorescence-activated cell sorting of specific cell types followed by RNA-seq (Warner *et al*, 2019). In addition, chromatin immunoprecipitation sequencing (ChIP-seq) targets of pan tissue-specific transcription factors, *pha-4* for the pharynx (Mango *et al*, 1994; Gaudet & Mango, 2002), *elt-1* for the skin (Page

*et al*, 1997; Gilleard & McGhee, 2001), *hlh-1*, *unc-120*, and *hnd-1* for the body wall muscle (Fukushige *et al*, 2006), *elt-2* for the intestine (Fukushige *et al*, 1998) were added to the tissue-specific gene lists. ChIP-seq targets of the above TFs (only embryo samples were considered) were obtained from the model organism Encyclopedia of Regulatory Networks (ModERN) database (Kudron *et al*, 2018).

### Enrichment of chromatin activity co-dynamic regions

For each cell from a clonal tissue lineage, all genomic positions exhibiting activated or silenced chromatin in that cell were identified and tested for whether these regions were enriched in specific clusters of chromatin activity co-dynamic regions. Enrichment analysis was performed by quantifying the ratio of observed-to-expected (O/E) overlap of genomic positions with the positions in each chromatin activity co-dynamic cluster, and the clusters with O/E score > 1.5 and a *Q*-value < 0.05 (Hypergeometric test) were considered to be enriched in a cell. If cellular enrichment of a chromatin activity co-dynamic cluster is observed in $\geq 50\%$ of the traced terminal cells of a clonal tissue lineage, the cluster was considered to be enriched in that lineage.

## Data availability

The datasets produced in this study are available in the following databases:

Image datasets: Biostudies, https://www.ebi.ac.uk/biostudies/studies/S-BIAD50

Analysis scripts: GitHub, https://github.com/IGDB-DuLab/Zhao-chromatin

Data visualization: http://dulab.genetics.ac.cn/chromatin-landscape

**Expanded View** for this article is available online.

## Acknowledgements

We thank Z. Bao for suggestions. This work was supported by grants from the "Strategic Priority Research Program" of the Chinese Academy of Sciences to Z.D. (XDB19000000), the National Natural Science Foundation of China to Z.D. (31722035 and 31771598), and the State Key Laboratory of Molecular Developmental Biology, China. Some strains were provided by the Caenorhabditis Genetics Center.

## Author contributions

ZD, ZZ, and RF conceived the project and designed the study. RF, WX, YK, YW, and XM conducted the experiments and generated the data. ZZ and ZD analyzed the data. ZD wrote the manuscript with input from all authors.

## Conflict of interest

The authors declare that they have no conflict of interest.

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
