## [Review Process File · Molecular Systems Biology]

Single-cell dynamics of chromatin activity during cell lineage differentiation in *C. elegans* embryos

Zhiguang Zhao, Rong Fan, Weina Xu, Yahui Kou, Yangyang Wang, Xuehua Ma, and Zhuo Du
DOI: [10.15252/msb.202010075](https://doi.org/10.15252/msb.202010075)

Corresponding author(s): *Zhuo Du (zdu@genetics.ac.cn)*

Review Timeline:

Submission Date:	20th Oct 20
Editorial Decision:	3rd Dec 20
Revision Received:	18th Jan 20
Editorial Decision:	26th Feb 21
Revision Received:	15th Mar 21
Editorial Decision:	22nd Mar 21
Revision Received:	22nd Mar 21
Accepted:	23rd Mar 21

Editor: Jingyi Hou

Transaction Report:

Thank you for submitting your work to Molecular Systems Biology. We have now heard back from the three reviewers who agreed to evaluate your manuscript. As you will see from the reports below, the reviewers acknowledge the potential interest of the study. They raise however a series of concerns, which we would ask you to address in a major revision.

Since the reviewers' recommendations are rather clear, there is no need to reiterate all the points listed below. Some of the key issues that would need to be addressed are the following:

- Many of the reviewers' concerns refer to the need to provide further clarifications and to improve the presentation of the study in order to make the data and the main conclusions easily accessible to the readers.
- In line with all three reviewers' comment, the conclusions need to be stated accurately and overstatements should be avoided.

All other issues raised by the reviewers need to be satisfactorily addressed as well. As you may already know, our editorial policy allows in principle a single round of major revision and it is therefore essential to provide responses to the reviewers' comments that are as complete as possible.

On a more editorial level, we would ask you to address the following issues.

REFEREE REPORTS

Reviewer #1:

Zhao et al. use eef-1A.1p::GFP NLS constructs integrated into 113 different locations along with lineage tracing during development to examine how local chromatin environment drives differential expression of the reporter during development in single embryonic cells in *C. elegans*. This is potentially both a very useful technique and resource, and provides information on the combined

features that regulate chromatin - histones, accessibility etc. They present evidence that chromatin states diversify in a lineage-dependent manner during development leading to important patterning such as anterior-posterior and left-right symmetry; and can contribute to cellular heterogeneity within tissues. Overall the work should be suitable for publication in Molecular Systems Biology with some revisions and clarifications.

My major concern that should be addressed before acceptance is that there is not sufficient discussion and analysis of the relative importance of global changes between insertions in expression levels that do not alter the underlying pattern vs changes that cause some cells to express the reporter more than others. It seems plausible that the reporter itself is biased to certain lineages or fates (such as intestine and skin), and that some of the "lineage differences" identified here are actually artifacts of lower-expressing lineages fluctuating above or below the detection threshold depending on global scaling of the reporter activity, compounded with measurement and biological noise. Several figures (1F, 1D, S1, S2F, S3, S5) as well as an examination of Table EV2 seem to support the idea that the overall pattern is fairly consistent between integrants. At a minimum, the authors should show how the distribution of Pearson correlation (as a proxy for "pattern" as opposed to "level") varies between integrants as opposed to between replicates for the same integrant (in other words a version of Fig. S2E but between integrants instead of between replicates). To be clear, I don't think even if global effects are dominant that this would render the paper uninteresting, in contrast, I think this might even be a more fundamental result if true. However regardless I think this issue must be carefully considered and addressed in the text.

Another suggestion is that a lot of the important details needed to understand and interpret results are buried in the methods section. The paper could also be improved with a little more clarity on what is being graphed in some of the figures and reducing the reach of some of the conclusions drawn. For example:

What GFP measurements are used for different analyses such as average GFP for that cell, instantaneous GFP expression calculated as $GFP(\text{daughter}) - GFP(\text{mother})$, etc?

Comparison with single cell transcriptome data:

Comparison to single cell expression: If the 500kb comes from the best correlation with the whole embryo time course it is worth mentioning in the main text.

Why were only 38 cells were found with unique identity (Figure 1I)? This seems like a small, and likely biased, subset.

Explanation of CAL and CAL divergence in the main text:

- Maybe FigS6B should be in the main figures early e.g. ~1G ?
- Figure 2C is hard to parse, needs much clearer labeling (what are the different panels??)

Transition points of CAL and inter/inter CAL divergence:

I was a bit unclear on if transition points are calculated for mother-daughter pairs only or mother-grand daughter and mother-great grand daughter also (as implied by Fig S9A/B).

Fig 2G clarify whether this is transition score overall vs. A/P transition score?

Can we find an easier term/way to refer to CDCA? The number of acronyms in the paper make it harder to read.

A generalized model that could differentiate between convergence and predetermination (mothers are less similar compared to intra-tissue daughters in convergence in contrast to mothers of L/R

daughters being similar to each other and the daughters in predetermination). If a cohesive model can be used in Fig 3K and 4H, that could be useful. Alternatively expected result of what 'convergence' would look like in contrast to the 'pre-determination' model for L/R symmetry.

4F - progenitors are only at the 50 cell and the rest are L-R pairs?

4G - what are the different bars?

More clarification of what is being shown in Fig. 5I

II. Conclusions that should be softened

Figure 1L - why pick magic number 5 with 67%? And the conclusion drawn from this 'Given that many embryonic cells have similar developmental fates, this means the cellular CAL is sufficient for defining cellular states at a sub-tissue level' is a bit inflated?

Figure 2C/S8A, B - 'Thus, cellular CAL diversifies as lineage unfolds and recapitulates the kinetics of lineage-coupled fate differentiation' - 'recapitulates kinetics' here is a bit overstated - as the data don't show that the CAL has similar information content to expression.

Furthermore, analysis of the cellular gene expression of L1 stage animals (Liu et al., 2009) consistently obtained an identical pattern (Appendix Fig S15A). This result thus reveals a chromatin basis for L-R functional asymmetry.' Identical pattern or just similar pattern?

Other suggestions

Some of the early figures (1,2,3) could be broken into 2 smaller figures to improve their readability.

Appendix Figure S10 - the most informative number of lineage groups could be used - and the heatmap combined with another figure?

Appendix Figure S1A: If possible to include in legend that green indicates the 113 strains examined and the 18 shown were PCR verified in this paper

In methods, the 'Analysis of the cellular dynamics and implications of CAL during cell lineage differentiation' section can be moved after the 'Comparison of position-effects on GFP with chromatin features' to keep with the flow of the paper. Similarly the Perturbation of writers of histone modifications could be moved ahead.

Figure 3C - would be better to show CAL divergence plots instead of Circos?

Page 4 "As compared to existing sequencing-based epigenomic approaches" - position effect analysis is not mutually exclusive with sequencing, really what this section needs is a brief mention of the advantages (e.g. cell ID/position) of imaging based vs sequencing approaches

Page 10 "Together, these results revealed a chromatin basis for the tissue-based convergence of regulatory states in cells originating from diverse lineages." - this isn't quite right. The authors showed that fate is correlated with chromatin activity convergence, not that chromatin controls convergent expression (could just as easily be the reverse or independent control by a third factor given the data presented).

Page 14 Charest et al (2020) showed that several other neurons thought previously to be symmetric have some molecular asymmetry, including AUA (listed here as an example of a symmetric neuron)

Existence and extent of TADs in *C. elegans* is controversial - indeed the paper cited identified "TAD-like" regions on the x-chromosome but not autosomes, but it is unclear if these are analogous to TADs seen in other organisms. More recent work has suggested that *C. elegans* TADS may exist on autosomes but be much smaller (on the order of 5-15 kb). This should be

addressed in the text of this section so it is clear what scale of "TADs" were analyzed here.

Reviewer #2:

The authors take a powerful approach to investigating the influence of chromatin environment on gene expression. They analysed the expression of the same reporter integrated into 113 sites at the level of single lineage-resolved cells during embryogenesis. They found abundant position effects on reporter expression level and used the expression data to infer a chromatin activity landscape (CAL) value for individual cells at each insertion site. The authors then used similarities and differences in CAL across lineages and cell types to make conclusions about the effects of chromatin regulation on fate patterning.

This is a good approach and the data on gene expression values appear to be of high quality. However, many analyses are difficult to understand, often because of insufficient information in the text, legends, and methods. In addition, many conclusions made are stronger than the data warrant. The authors need to be more careful and considered in their conclusions and to explain their analyses more clearly. With considerable re-writing, this paper could be of wide interest.

Below I give examples of issues that the author need to improve.

1. Most analyses in the paper rely on their CAL metric, but this metric is not clearly explained. As far as I can see, CAL is a metric based on the expression level of the same reporter gene (ubiquitous *eef-1* promoter driving GFP) in embryonic cells when integrated into 113 genomic sites. In the first part of the results, the authors need to include much more explanation of what CAL is. In the text, they simply say that their assays "allowed a cell-by-cell integration of GFP expression levels at different genomic positions in lineage-equivalent cells (Fig 1F). This enabled us to construct a chromatin activity landscape (CAL) in each lineaged cell (Fig 1G, Table EV3)." The method for calculating CAL is unclear. Is it simply the expression level? For ABalpaapaa in SS343 (Chr I, 0.53), there are expression values for two embryos: .650484648 and 2.968127764. The CAL for this cell is 2.2318772. How was this was derived? The authors also need to explicitly explain that they use gene expression measurements to infer effects of the chromatin landscape and that they will use CAL values as a proxy for chromatin state.

2. Some additional basic information about the reporter is needed. For example, was the *eef-1* reporter indeed ubiquitously expressed as expected, at least in some integration positions? In the methods, they refer to S2F to support that the *eef-1A* reporter is expressed in most cells from the 350 cell stage and onwards. Indeed, nearly every cell does appear to show expression from at least one integration site, but in their analyses they only considered cells where expression was seen in 60% of the embryos: "only cells in which GFP is expressed in more than 60% of the embryos were considered expressing cells" implying non-ubiquitous expression. This cutoff and its rational needs to be discussed and justified in the main text. From first principles, it seems to me that cells that show a large divergence in expression would be highly informative. In addition, the authors should explain what types of expression changes they observed. Was expression from some insertions completely lost in specific cells and/or lineages, or were effects more often changes in levels?

3. In many places, the authors use strong language for their conclusion coupled with vague statements. The nature and limits of their conclusions need to be more precisely and accurately made. Clarifying what CAL means early on in the paper will help. Examples: p. 9 we "found that the inferred CAL transitions immediately result in significantly larger transcriptome divergences" Here they strongly conclude a cause and effect relationship. They have shown a correlation, not a causation. At the end of this paragraph, they write "Thus, chromatin transitions predict anterior-posterior asymmetry during lineage progression." This should be "inferred chromatin transitions." On p. 13, they write "This result thus reveals a chromatin basis for L-R functional asymmetry." Here it would be more appropriate to change "reveals" to "supports." Similar issues are found throughout the paper.

4. I could not follow the first paragraph of the section "Chromatin diversification predicts lineage-coupled fate determination."

5. I didn't understand their concluding sentence of the first paragraph on p. 12 " Intriguingly, analysis of lineage-resolved expression data from cells of L1 stage animals (Liu, Long et al., 2009) demonstrated that the lineage effects not only result in gene expression heterogeneity but also are stably maintained after hatching (Appendix Fig S13D)."

6. In the discussion " Through a multidimensional analysis of the lineage-resolved chromatin landscape, we reveal that the regulation of lineage commitment, anterior-posterior asymmetry, tissue fate specification, cell heterogeneity, and bilateral symmetry establishment are readily inferable from cellular chromatin." implies that the authors have specific information on cellular chromatin state.

7. The legends need much more information in order to understand the figures. e.g, Figure 1G "CAL of representative embryonic cells. There is no explanation of colors. 1H on histone modifications is impossible to understand without reading the methods. Most figure legends need more information.

8. In the explanation of how GFP was quantified in the methods, the authors say that GFP intensities of the same cell at multiple time points were averaged. How many time points per cell? What is the range?

9. Which TADs were used from Crane et al 2015?

10. Images in Figure S5 B and C are hard to see. What statistical test was performed when comparing wt and mutant data?

11. Ho et al, 2014 and Evans et al, 2016 derived chromatin states from *C. elegans* embryo histone modification data. Are these states informative with respect to the CAL values? These data may allow the authors to infer the type of chromatin that may be driving the observed relationships.

Reviewer #3:

Major Revision Recommended

In this manuscript, Zhao et al. present a study of position effect variegation (PEV) on reporter gene

expression using live imaging during *C. elegans* embryogenesis. The study of PEV on gene expression during differentiation is important, timely, and interesting, and the scope of the study (hundreds of different genomic sites investigated) is technically impressive. The most convincing findings were that individual lineages may have characteristic activity profiles (by their CAL metric), and that tissues arising from diverse lineages have large heterogeneity in CAL divergence among their cells.

However, I am not convinced by many of the other conclusions drawn by the authors, which I detail below. Most importantly, the authors should temper their conclusions because this is a reporter-based system with a strong promoter and it is not known if the conclusions will hold true for endogenous sequences. In fact, from Figure S4F, the overall correlation with endogenous gene expression appears poor. This result casts some doubt on the physiological relevance of the reporter data. One practical solution may be to only consider reporter data from an integrated location where there was high correlation with nearby unmodified endogenous genes.

Throughout the manuscript, the authors should not assume that chromatin changes precede gene expression and fate changes as the embryo develops, because this has not been proven. They should avoid statements to this end (e.g. chromatin regulation of cell differentiation etc), and instead state their apparent correlations plainly.

General comments:

The language used in the title and abstract is vague. These elements should be concise and clearly written, and avoid new unexplained terminology.

Not all figure panels are referenced in the main text, leaving the reader wondering what conclusions can be drawn from these missing panels.

Figure legends are under-described, especially in the supplement. They should explain how each experiment was done briefly but clearly and what conclusions can be directly drawn from each.

New terminology like CAL and CDCA which are critical to the message of the manuscript need to be thoroughly and clearly described in the main text.

The code used to analyze the data and produce the figures should be deposited in an appropriate online location.

Specific comments:

Pg 5 "to explore chromatin regulation of cell differentiation in *C. elegans*," This is very vague, and the data is only correlative, so the authors should be careful not to imply causation without evidence. This wording implies chromatin regulates cell differentiation.

What they are doing is cataloging the expression levels of a reporter in ~100 different locations using previously generated strains, not measuring cell differentiation. They can say that cells of different lineages and developmental ages have different reporter expression patterns (or chromatin activity landscapes to use the authors' term).

Pg 5 "These findings will contribute to a systems-level mechanistic understanding of how chromatin regulates cell lineage differentiation in a metazoan embryo across cell lineages, tissue types, and symmetric morphological organization."

I disagree with this statement. The data in the manuscript does not support this claim. There are no

proofs in the manuscript of chromatin regulating lineage differentiation.

Pg 6 "In total, we measured GFP expression in 268 embryos to quantify chromatin activity at 113 genomic positions"

In main text explicitly state how many times each strain was recorded and analyzed. It appears that only duplicates were performed (from Figure S2). From the duplicate data in Figure S2, the correlation can range from 0 to 1. The authors should discuss this variation in the main text. These experiments form the basis of the entire manuscript and should be conducted in biological triplicates to be convincing. The replicates should be added to Figure S3 so that readers can visually assess the variability.

All images of the GFP reporters should have the nuclear mCherry marker side-by-side as a control.

It is perplexing that reporter expression correlates with histone modification status of the whole embryo (as in mutant and RNAi experiments Figure S5) but not the expression levels of nearby genes (Figure 1I, S4). I do not agree with statements in the text such as "the measured chromatin activity is highly concordant with endogenous gene expression in the same cells" (pg 7) because the data show low Pearson R values 0.3-0.6. It appears that the reporter activity level at its various integrated locations is a poor predictor of endogenous gene activity.

The term CAL and what data it encompasses should be more clearly stated in the main text.

I am not convinced by the statement that "Given that many embryonic cells have similar developmental fates, this means the cellular CAL is sufficient for defining cellular states at a sub-tissue level." (pg 7) How does the data in Figure 1L prove this?

The authors state that "chromatin transitions predict anterior-posterior asymmetry during lineage progression", however I see no data showing CAL divergence predicts the fate of anterior cells or posterior cells in the embryo.

Figure S10 is quite interesting. Can this data (fraction of positions with distinct on/off states) be normalized to lineage distance? This will help to show which results are expected/surprising, and both cases should be discussed.

The authors should avoid statements such as the underlined: "Collectively, these findings support that cellular chromatin diversifies considerably during early lineage progression and systematically predicts lineage-coupled fate differentiation, including the global lineage-dependent diversification of cell fates, anterior-posterior fate asymmetry, and the establishment of lineage-specific fates." (pg 9) I see no evidence at this point in the manuscript that a particular CAL can predict a particular fate. It is unclear how different CALs are between different lineages. The authors should describe more clearly how CAL divergence is calculated in the main text.

The data in Fig S14 is interesting but it is impossible to see the cell IDs in the matrix. This panel should be much bigger. The authors should discuss both unexpected and expected results.

The L-R symmetry section is difficult to understand or find compelling. It is unclear at what point in development CAL divergences are reduced in L-R cell pairs and how this compares to transcriptional divergence.

Can the authors compare the data derived from the Pnhr reporter strains to the Peef-1A.1 strains?

Do close by integration sites produce similar CALs? This could help to show the robustness of CAL as a metric.

The methodology for CDCA calculations is unclear and therefore the relevance of this data is not apparent. The authors should avoid vague terminology like "inter-cell dynamics".

Response to reviewers

We thank all reviewers for the time they spent evaluating our work and for the constructive suggestions made in the interest of improving the study. Following the suggestions, we have significantly revised the text and figures of the manuscript (all changes are highlighted in blue in the manuscript file). Below, we provided point-by-point responses (bold blue) to all concerns and suggestions raised by the reviewers.

Reviewer #1:

Zhao et al. use eef-1A.1p::GFP NLS constructs integrated into 113 different locations along with lineage tracing during development to examine how local chromatin environment drives differential expression of the reporter during development in single embryonic cells in *C. elegans*. This is potentially both a very useful technique and resource, and provides information on the combined features that regulate chromatin - histones, accessibility etc. They present evidence that chromatin states diversify in a lineage-dependent manner during development leading to important patterning such as anterior-posterior and left-right symmetry; and can contribute to cellular heterogeneity within tissues. Overall the work should be suitable for publication in *Molecular Systems Biology* with some revisions and clarifications.

Response: We thank the reviewer for the kind words and for the accurate summary of our work. As you can see in the revised manuscript, we have significantly revised the text and figures to present our findings more clearly and accurately.

My major concern that should be addressed before acceptance is that there is not sufficient discussion and analysis of the relative importance of global changes between insertions in expression levels that do not alter the underlying pattern vs changes that cause some cells to express the reporter more than others. It seems plausible that the reporter itself is biased to certain lineages or fates (such as intestine and skin), and that some of the "lineage differences" identified here are actually artifacts of lower-expressing lineages fluctuating above or below the detection threshold depending on global scaling of the reporter activity, compounded with measurement and biological noise. Several figures (1F, 1D, S1, S2F, S3, S5) as well as an examination of Table EV2 seem to support the idea that the overall pattern is fairly consistent between integrants. At a minimum, the authors should show how the distribution of pearson correlation (as a proxy for "pattern" as opposed to "level") varies between integrants as opposed to between replicates for the same integrant (in other words a version of Fig. S2E but between integrants instead of between replicates). To be clear, I don't think even if global effects are dominant that this would render the paper uninteresting, in contrast, I think this might even be a more fundamental result if true. However regardless I think this issue must be carefully considered and addressed in the text.

Response:

Thank you for raising this concern. In the revised manuscript, we have systematically addressed this concern in the Results section (see Cellular chromatin is dynamic and informative for defining cellular states), and also added several figure panels in Fig 2.

First, as suggested by the reviewer, we directly compared GFP expression divergence (measured as Euclidian distance) between replicates and between different positions. Both quantitative levels and on/off expression patterns showed that the GFP expression divergences between different integration sites are significantly larger than those between replicates at a

given position (Fig 2B). This result suggests that the *eef-1A.1* promoter sequence does not significantly dominate the position-effects on GFP expression.

Second, we took another approach to assess potential bias of the *eef-1A.1* promoter. If the promoter sequence is biased towards certain cells, we would expect GFP to be constitutively expressed or not expressed in those cells regardless of integration site. However, analysis of the binary expression pattern of GFP showed that only in a small number of cells was GFP constitutively expressed or silenced across all integration sites (Fig 2D). Specifically, in less than 10% of the cells ($n = 35$), GFP was constitutively expressed (expressed at >80% of integration sites), and in less than 5% of cells ($n = 15$), GFP was constitutively silenced (expressed at <20% of integration sites). This suggests that when changing chromosome locations, the *eef-1A.1* promoter could either be on or off, and the promoter does not exhibit strong expression bias in many cells.

Third, as the reviewer pointed out, we can not completely rule out the possibility that the promoter is biased towards a small number of cells. For example, this reporter is more frequently expressed in certain cells from the C and E lineages than in other cells. It is also plausible that chromatin states could be more active and supportive of gene expression in

certain cells. Nevertheless, we think such a small fraction of cells is unlikely to affect the major conclusions of this study.

Another suggestion is that a lot of the important details needed to understand and interpret results are buried in the methods section. The paper could also be improved with a little more clarity on what is being graphed in some of the figures and reducing the reach of some of the conclusions drawn.

Response: First, we have significantly expanded the Results section by incorporating all relevant and essential information originally described in the Methods. We believe this modification will improve the readability. Second, we significantly expanded the figure legends to include more details on what is analyzed and presented. Finally, we systematically moderated our tone on the implications of our findings. Specifically, we removed the phrase “chromatin regulation of ...” throughout the manuscript and changed it to “chromatin dynamics during/accompanying ...” or “chromatin dynamics correlate with ...”

For example:

What GFP measurements are used for different analyses such as average GFP for that cell, instantaneous GFP expression calculated as $GFP(\text{daughter}) - GFP(\text{mother})$, etc?

Response: We used instantaneous GFP expression for all analyses. We have now pointed this out in explicitly the Methods: “Instantaneous GFP expression levels were used to represent chromatin activity.”

Comparison with single cell transcriptome data:

Comparison to single cell expression: If the 500kb comes from the best correlation with the whole embryo time course it is worth mentioning in the main text.

Why were only 38 cells were found with unique identity (Figure 1I)? This seems like a small, and likely biased, subset.

Response:

1) In the revised manuscript, we have now included this information in the Results section: “We observed a considerable correlation when interval size was extended to over 100 kb, with the 500-kb interval yielding the strongest correlation (Fig EV2L, right, $R = 0.57$, $P = 3.76E-11$).”

2) In the original study, only a small fraction of cells (~8%) had a unique lineage identity assigned (Packer *et al.*, 2019). Thus, the 38 cells comprise all cells that we can use without bias. This limitation exists because only a limited number of marker genes were used to determine cell identity and because many cells with distinct lineage identities may have very similar transcriptomes (for example, the left-right symmetric cells). To include more examples, we have extended our analysis to those cells that were assigned two possible identities ($n = 247$), for which we obtained a similar result. This new data is added in Fig EV2M.

Explanation of CAL and CAL divergence in the main text:

- Maybe FigS6B should be in the main figures early e.g. ~1G ?
- Figure 2C is hard to parse, needs much clearer labeling (what are the different panels??)

Response:

1) We have now added explanations of chromatin activity landscape and divergence in the main text as follows:

P7 - “Using GFP expression levels at different genomic positions as an indicator of chromatin activity, we constructed the distribution of chromatin activity across 113 genomic positions (termed the chromatin activity landscape) for all lineage-traced cells.”

P13 – “Chromatin divergence was measured as the Euclidean distance between chromatin activity (GFP expression) across all integration sites in a cell.”

2) We have moved Fig S6B to Fig 1G.

3) Each panel corresponds to the result for cells at a certain lineage distance. We have added lineage distance labeling in these figure panels (Fig 3D).

Transition points of CAL and inter/inter CAL divergence:

I was a bit unclear on if transition points are calculated for mother-daughter pairs only or mother-grand daughter and mother-great grand daughter also (as implied by Fig S9A/B).

Fig 2G clarify whether this is transition score overall vs. A/P transition score?

Response:

(1) We did not directly compare a mother cell to its descendants. Instead, we applied a retrospective approach to infer chromatin transitions by comparing chromatin landscapes observed in all traced terminal cells. We have added this information (previously described in

the Methods) to the Results section for clarity. “A cell division is defined as a transition point for the chromatin landscape if the chromatin divergences between terminal cells generated by different daughter cells (inter-divergence) are significantly higher than those between terminal cells generated by the same daughter cell (intra-divergence) (Fig EV3D).”

(2) The scatter plot compares the chromatin transition score (x-axis) and A/P fate divergence score (y-axis) associated with each of the 90 early cell divisions. Detailed information can be found in Table EV6.

Can we find an easier term/way to refer to CDCA? The number of acronyms in the paper make it harder to read.

Response: Thank you for the suggestion. (1) We have used the term “chromatin co-dynamic regions” to replace “CDCA”. (2) We have also used the spelled-out version of “chromatin activity landscape” instead of “CAL” to reduce the number of acronyms.

A generalized model that could differentiate between convergence and predetermination (mothers are less similar compared to intra-tissue daughters in convergence in contrast to mothers of L/R daughters being similar to each other and the daughters in predetermination). If a cohesive model can be used in Fig 3K and 4H, that could be useful. Alternatively expected result of what 'convergence' would look like in contrast to the 'pre-determination' model for L/R symmetry.

Response: Thank you for this helpful comment. We have used a cohesive model in Fig 5I to compare the convergence and predetermination models.

4F - progenitors are only at the 50 cell and the rest are L-R pairs?

4G - what are the different bars?

Response:

(1) We meant to take the 50 cell stage as an example to illustrate that chromatin divergences between L-R progenitor cells at this early stage are similarly low as in the 350-cell stage. L-R progenitor cells exist in multiple developmental stages, especially before the 350-cell stage. We have revised this figure panel to directly compare chromatin divergence between L-R progenitor cells at different developmental stages and between L-R cells at the 350-cell stage.

(2) Each bar plot shows the *Pnhr-2::GFP* expression levels across 13 genomic positions in a cell. We have added axis labels in the figure.

More clarification of what is being shown in Fig. 5I

Response: We have added more description in the legend of Fig 5I (now Fig 6J):

“Clusters of chromatin co-dynamic regions (barcodes) are enriched in all genomic regions that exhibit active (purple lines) or silent (cyan lines) chromatin states in individual cells from corresponding clonal tissue lineages (colored circles, coded according to tissue type).”

II. Conclusions that should be softened

Figure 1L - why pick magic number 5 with 67%? And the conclusion drawn from this 'Given that many embryonic cells have similar developmental fates, this means the cellular CAL is sufficient for defining cellular states at a sub-tissue level' is a bit inflated?

Response: We apologize for this misunderstanding. We have significantly revised this section to better illustrate the point that cellular chromatin landscapes could distinguish cells at the sub-tissue level (see below). The selection of five as the cut-off is a practical choice because this number would give someone multiple options when selecting one site by which to distinguish two cells. It should be noted that when determining the number of integration sites exhibiting different on/off expression patterns between two cells, those sites that showed variable expression among experimental replicates were not considered (we have added this important information in the figure legend).

P12 - “We finally examined to what extent the cellular chromatin landscape can distinguish individual cells. We compared GFP expression patterns across cells and calculated for each pair-wise comparison the number of integration sites at which GFP expression status is distinct. Intriguingly, for most cell-cell comparisons, the binary GFP expression at many integration sites was distinct, and at a considerable number of integration sites, the expression status can distinguish a cell from many other cells (Fig 2G). On average, cellular chromatin distinguished a cell from 79.4% of other cells if the on/off state of chromatin activity at five or more genomic positions was distinct in a cell-cell comparison. The tissue-level analysis further showed that cellular chromatin not only distinguished a cell from a large fraction of cells belonging to a different tissue type but also distinguished it from a considerable fraction of cells of the same tissue type (Fig 2H). Thus, the cellular chromatin landscape provides rich information for distinguishing cells at a sub-tissue level.”

Figure 2C/S8A, B - 'Thus, cellular CAL diversifies as lineage unfolds and recapitulates the kinetics of lineage-coupled fate differentiation' - 'recapitulates kinetics' here is a bit overstated - as the data don't show that the CAL has similar information content to expression.

Response: We have changed this statement to “Thus, chromatin dynamics during lineage progression correlate with lineage-coupled cell differentiation.”

Furthermore, analysis of the cellular gene expression of L1 stage animals(Liu et al., 2009) consistently obtained an identical pattern (Appendix Fig S15A). This result thus reveals a chromatin basis for L-R functional asymmetry.' Identical pattern or just similar pattern?

Response: Sorry for the confusion. We meant a similar pattern. We have revised the statement as follows:

“Furthermore, this pattern was also evident in cellular gene expression data from L1 stage animals (Liu *et al.*, 2009), in which gene expression divergences between L-R cells were significantly lower than those between other cells having matched lineage distance within a tissue (Appendix Fig S5B).”

Other suggestions

Some of the early figures (1,2,3) could be broken into 2 smaller figures to improve their readability.

Response: Thank you for the suggestion. We have significantly revised the figures and split the old Figure 1 into two figures. In addition, some panels originally in the supplementary figures are now incorporated into main figures.

Appendix Figure S10 - the most informative number of lineage groups could be used - and the heatmap combined with another figure?

Response: We have now only used the result for 50 lineage groups and combined this panel with another figure (Fig EV3).

Appendix Figure S1A: If possible to include in legend that green indicates the 113 strains examined and the 18 shown were PCR verified in this paper

Response: We have revised the figure legend as suggested. Thank you.

In methods, the 'Analysis of the cellular dynamics and implications of CAL during cell lineage differentiation' section can be moved after the 'Comparison of position-effects on GFP with chromatin features' to keep with the flow of the paper. Similarly the Perturbation of writers of histone modifications could be moved ahead.

Response: Thank you for catching this. We have adjusted the order accordingly.

Figure 3C - would be better to show CAL divergence plots instead of Circos?

Response: We meant to use the Circos plots to illustrate the point that cells of the same tissue type exhibit similar chromatin landscapes because these would be more intuitive than a divergence plot. Following the suggestion, we have added a divergence plot and statistics to illustrate the convergence of the chromatin landscapes.

Page 4 "As compared to existing sequencing-based epigenomic approaches" - position effect analysis is not mutually exclusive with sequencing, really what this section needs is a brief mention of the advantages (e.g. cell ID/position) of imaging based vs sequencing approaches

Response: Thanks for the suggestion. A brief discussion has been added to mention the advantages of the imaging-based approach.

Page 10 "Together, these results revealed a chromatin basis for the tissue-based convergence of regulatory states in cells originating from diverse lineages." - this isn't quite right. The authors showed that fate is correlated with chromatin activity convergence, not that chromatin controls convergent expression (could just as easily be the reverse or independent control by a third factor given the data presented).

Response: Thank you for this suggestion. We have now softened our statements throughout the manuscript. Specifically, the words "correlate" or "accompanying" are used to replace "chromatin basis" or "chromatin regulation".

Page 14 Charest et al (2020) showed that several other neurons thought previously to be symmetric have some molecular asymmetry, including AUA (listed here as an example of a symmetric neuron)

Response:

We have added this new reference, thanks. Please note that we have moved this section into the Discussion.

“In rare cases, L-R symmetric cells exhibit functional asymmetry (Hobert, 2014). For example, a pair of morphologically and positionally symmetric neurons (ASEL/R) show differential chemosensory capacities (Pierce-Shimomura *et al*, 2001); this asymmetry is known to be primed in early progenitor cells through a chromatin-based mechanism (Cochella & Hobert, 2012). Such predetermined asymmetry was not evident in the chromatin landscapes of ASEL/R progenitor cells (ABalpppppp/ABpraaapp), which exhibit a low chromatin divergence comparable to other L/R symmetric cells/progenitors from the same symmetric lineages (ABalpp/ABpraa) that do not have known functional asymmetry ($P = 0.2$). In addition to the asymmetric ASEL/R, the ABalpppppp/ABpraaapp progenitors also produce two pairs of L-R neuronal cells, including the AUAL/R cells, which have been recently shown to exhibit molecular asymmetry (differential expression of *C32C4.16*) (Charest *et al*, 2020), and the ASJL/R cells, in which no functional asymmetry has been observed. It is thus possible that regulation of L-R asymmetry may hinge on specific chromatin loci rather than the chromatin landscape as a whole. Indeed, differential chromatin decompaction of the *Isy-6* miRNA locus has been shown essential for priming ASEL/R asymmetry (Cochella & Hobert, 2012).”

Existence and extent of TADs in *C. elegans* is controversial - indeed the paper cited identified "TAD-like" regions on the x-chromosome but not autosomes, but it is unclear if these are analogous to TADs

seen in other organisms. More recent work has suggested that *C. elegans* TADS may exist on autosomes but be much smaller (on the order of 5-15 kb). This should be addressed in the text of this section so it is clear what scale of "TADs" were analyzed here.

Response: Thank you for pointing this out. We used the regions between the TAD boundaries defined by the authors to represent TADs (Crane et al., 2015). However, as this reviewer mentioned, the existence and function of TAD-like structures in *C. elegans* autosomes are unclear; we thus removed all results related to TADs. We apologize for the misunderstanding.

Reviewer #2:

The authors take a powerful approach to investigating the influence of chromatin environment on gene expression. They analysed the expression of the same reporter integrated into 113 sites at the level of single lineage-resolved cells during embryogenesis. They found abundant position effects on reporter expression level and used the expression data to infer a chromatin activity landscape (CAL) value for individual cells at each insertion site. The authors then used similarities and differences in CAL across lineages and cell types to make conclusions about the effects of chromatin regulation on fate patterning.

This is a good approach and the data on gene expression values appear to be of high quality. However, many analyses are difficult to understand, often because of insufficient information in the text, legends, and methods. In addition, many conclusions made are stronger than the data warrant. The authors need to be more careful and considered in their conclusions and to explain their analyses more clearly. With considerable re-writing, this paper could be of wide interest.

Response: We thank the reviewer for these positive comments about the approach and findings and for the valuable suggestions to improve the manuscript. We have significantly revised and expanded the manuscript by providing necessary background information and analysis details in the main text and figure legends. Furthermore, we have systematically softened our language when describing the conclusions to accurately represent the data. Specifically, the words “correlate” or “accompanying” are used to replace “chromatin basis” or “chromatin regulation”.

Below I give examples of issues that the author need to improve.

1. Most analyses in the paper rely on their CAL metric, but this metric is not clearly explained. As far as I can see, CAL is a metric based on the expression level of the same reporter gene (ubiquitous eef-1

promoter driving GFP) in embryonic cells when integrated into 113 genomic sites. In the first part of the results, the authors need to include much more explanation of what CAL is. In the text, they simply say that their assays "allowed a cell-by-cell integration of GFP expression levels at different genomic positions in lineage-equivalent cells (Fig 1F). This enabled us to construct a chromatin activity landscape (CAL) in each lineaged cell (Fig 1G, Table EV3)." The method for calculating CAL is unclear. Is it simply the expression level? For ABalpaapaa in SS343 (Chr I, 0.53), there are expression values for two embryos: .650484648 and 2.968127764. The CAL for this cell is 2.2318772. How was this was derived? The authors also need to explicitly explain that they use gene expression measurements to infer effects of the chromatin landscape and that they will use CAL values as a proxy for chromatin state.

Response:

(1) It is correct that the chromatin activity landscape described in this study refers to the expression levels of *Peef-1A.1::GFP* across 113 integration sites in a cell. We have added an explanation of this term in the Results section as follows:

“*C. elegans* embryogenesis follows a fixed cell lineage pattern and generates the same set of differentiated cells in each embryo, making embryos entirely comparable at single-cell resolution (Sulston *et al*, 1983). Thus, although the cellular GFP expression at each integration site was assayed in individual embryos (Appendix Fig S1), the invariant lineage allowed a cell-by-cell integration of GFP expression levels at different genomic positions in lineage-equivalent cells (Fig 1F). Using GFP expression levels at different genomic positions as an indicator of chromatin activity, we constructed the distribution of chromatin activity across 113 genomic positions (termed the chromatin activity landscape) for all lineage-traced cells (Fig 1F-H and Table EV3). For multiple replicates (range 2-8) of GFP expression at the same integration site, only those expressed in more than 60% of replicates were

considered as being expressed, and levels were averaged to represent the consensus chromatin activity.”

(2) Chromatin landscapes were constructed simply using the GFP expression level. The discrepancy this reviewer pointed out is caused by the order of the data transformations used to present the results. When we show cellular GFP expression in individual embryos, the expression levels were $\log_2(X+1)$ transformed. However, when integrating expression for multiple embryos, the untransformed values were averaged and then $\log_2(X+1)$ transformed. In other words, the untransformed values for the two embryos were 0.570 and 6.825, for which averaging and $\log_2(X+1)$ transformation yielded 2.232. Thank you for catching this discrepancy, and we have now revised the Methods section for clarity.

“...and the instantaneous expression levels (untransformed) were averaged and $\log_2(X+1)$ transformed to represent consensus chromatin activity at a genomic location in a cell.”

2. Some additional basic information about the reporter is needed. For example, was the eef-1 reporter indeed ubiquitously expressed as expected, at least in some integration positions? In the methods, they refer to S2F to support that the eef-1A reporter is expressed in most cells from the 350 cell stage and onwards. Indeed, nearly every cell does appear to show expression from at least one integration site, but in their analyses they only considered cells where expression was seen in 60% of the embryos: "only cells in which GFP is expressed in more than 60% of the embryos were considered expressing cells" implying non-ubiquitous expression. This cutoff and its rationale needs to be discussed and justified in the main text. From first principles, it seems to me that cells that show a large divergence in expression would be highly informative. In addition, the authors should explain what types of expression changes they observed. Was expression from some insertions completely lost in specific cells and/or lineages, or were effects more often changes in levels?

Response:

Thank you for raising this concern.

(1) Yes, when integrated into certain genomic positions, the reporter is expressed in most traced terminal cells, supporting that the promoter at least has the potential to be ubiquitously expressed. In addition, in all cells, we found that the reporter is expressed at a substantial fraction of all integration sites. These results support the ubiquitous nature of the *eef-1A.1* promoter.

(2) "only cells in which GFP is expressed in more than 60% of the embryos were considered expressing cells" — This statement relates to dealing with the variability in GFP expression among experimental replicates when integrated into the same position. The 60% number refers to experimental replicates at an integration site, rather than across different integration sites. In other words, if the majority (60%) of experimental replicates showed expression, we treated the reporter as being expressed in a cell when integrated at that position. Because 2-3 embryos were analyzed for each integration strain in most cases, 60% is a reasonable choice. As you can see in Table EV4, all of the 364 traced terminal cells were considered, regardless of the expression divergence across different positions observed when constructing the chromatin landscape. We have revised the Methods section to make this point more clear.

“Because GFP expression at each integration site was quantified for multiple embryos (ranging from 2 to 8), expression levels were integrated across experimental replicates. In this integration, only cases in which GFP was expressed (with a value >0) in more than 60% of replicates were considered to be expressed, and the instantaneous expression levels (untransformed) were averaged and $\log_2(X+1)$ transformed to represent consensus chromatin activity at a genomic location in a cell. Otherwise, the chromatin activity was set to zero.”

(3) We have added a section in the Results to provide additional information on reporter expression. In short, both on/off and quantitative expression changes were observed. In Table EV4, a value of 0 indicates no expression.

P10 - “Having established that the inferred chromatin landscape is reliable, we next determined the extent to which chromatin activity changes across positions and cells. We first examined whether the expression of *Peef-1A.1::GFP* across genomic positions changes considerably, taking the expression variability at each position into consideration (Fig 2A). Both quantitative levels and on/off expression patterns showed that the GFP expression divergences between different integration sites were significantly larger than those between replicates at a given position (Fig 2B). This result suggests that the *eef-1A.1* promoter sequence does not significantly dominate the position-effects on GFP expression.

While GFP expression was generally consistent between experiment replicates (Fig EV1G), highly variable expression was observed in certain cells at certain genomic positions (Appendix Fig S2). Because only a small number of experimental replicates were performed, we selected four insertion strains exhibiting high variability and quantified GFP expression in more embryos. The correlation of GFP expression between replicates at these positions remained low (Appendix Fig S2), suggesting chromatin states in certain regions could be flexible. Indeed, chromatin state/activity has been previously shown to be variable and stochastic at certain positions (Angermueller *et al*, 2016). Furthermore, when studying the position-effect variegation phenotype of fly eye color, epigenetic silencing of the *white* gene has been shown to be highly stochastic, causing a variegated red and white color phenotype (Timms *et al.*, 2016).

Qualitatively, chromatin activity is highly dynamic across the genome and among cells. This was determined by comparing cellular and positional dynamics of the on/off status of GFP expression. At each integration site, GFP was expressed in a proportion of the 364 cells

(Fig 2C, median = 60%); at only seven positions, GFP was active (n = 1) or silenced (n = 6) in all cells. It suggests that chromatin activity at most genomic positions exhibited cell-specificity. In each cell, GFP was expressed only when it had been integrated into a fraction of genomic positions (Fig 2D, median = 67%), suggesting that chromatin activity in a cell is positionally specific. In only a small number of cells, the GFP was constitutively expressed or not expressed across all integration sites. Specifically, in less than 10% of the cells (n = 35), GFP was constitutively expressed (expressed at >80% of the integration sites), and in less than 5% of the cells (n = 15), GFP was constitutively silenced (expressed at <20% of the integration sites). It suggests that when changing the chromosome location, *eef-1A.1* promoter in a cell could either be on or off, and that the promoter does not exhibit strong expression bias to many cells.

Using quantitative expression data, we examined whether the positional or cellular dynamics of chromatin could confer rich information. If it does, then the expression levels would be distributed evenly across different magnitude categories. Adopting the concept of information content (IC) that estimates the amount of information conveyed by variables, we first estimated the theoretical maximum positional and cellular IC giving the expression range of GFP expression levels and then compared the observed IC to the theoretical maximum value (Materials and Methods). Significantly, the positional and cellular IC of the chromatin landscape reached 75% and 69% of the maximum, respectively (Fig 2E and F), indicating the information conferred by chromatin is rich.”

3. In many places, the authors use strong language for their conclusion coupled with vague statements. The nature and limits of their conclusions need to be more precisely and accurately made. Clarifying what CAL means early on in the paper will help. Examples: p. 9 we "found that the inferred CAL transitions immediately result in significantly larger transcriptome divergences" Here they strongly

conclude a cause and effect relationship. They have shown a correlation, not a causation. At the end of this paragraph, they write "Thus, chromatin transitions predict anterior-posterior asymmetry during lineage progression." This should be "inferred chromatin transitions." On p. 13, they write "This result thus reveals a chromatin basis for L-R functional asymmetry." Here it would be more appropriate to change "reveals" to "supports." Similar issues are found throughout the paper.

Response: Thank you for pointing this out. We have systematically revised or removed overstatements throughout the manuscript to acknowledge that correlation rather than causality was observed between chromatin dynamics and cell differentiation. In particular, we have revised the three statements the reviewer pointed out as follows:

“Indeed, transcriptome divergences between daughter cells showing chromatin transitions were significantly larger than between those without transitions (Fig EV3F).”

“Thus, the inferred chromatin transitions correlate with anterior-posterior asymmetry during lineage progression.”

The third statement is removed.

4. I could not follow the first paragraph of the section "Chromatin diversification predicts lineage-coupled fate determination."

Response: Sorry for the confusion. We have significantly expanded this paragraph to explain the reasoning behind the analysis.

“Cell differentiation accompanies lineage progression. The lineage-based mechanism plays a crucial role in initiating cell differentiation by assigning distinct fates to progenitor cells in a lineage-dependent manner, hence diversifying cell fates (Labouesse & Mango, 1999). Accordingly, it is natural to ask whether the cellular chromatin landscape diversifies during lineage progression and indicates lineage-coupled fate differentiation. We first quantified

chromatin divergence as a function of the lineage relationship between 364 traced terminal cells. Chromatin divergence was measured as the Euclidean distance between chromatin activity (GFP expression levels) across all integration sites in a cell (Fig 3A). Lineage relationship was quantified as cell lineage distance, which was defined as the total number of cell divisions separating cells from their lowest common ancestor (Fig EV3A). In the majority of cases, higher chromatin divergences were observed between cells with a large lineage distance and, globally, chromatin divergence increased progressively with cell lineage distance (Fig 3B and C). Thus, in general, chromatin diversifies gradually across cells during lineage progression.

We next analyzed the lineage-coupled kinetics of cell differentiation by measuring how cell fates change as a function of the lineage distance between cells. Based on the lineage tree structure and tissue types of all terminally differentiated cells, we retrospectively defined progenitor cell fate as the combinatorial pattern of tissue types produced by each cell and quantified the fate difference between cells (Materials and Methods and Fig EV3B). This analysis showed that, generally, as the cell lineage unfolds, cells differentiate progressively. The fate divergences between cells were proportional to their lineage distances at different developmental stages, similar to what was observed with chromatin divergences (Fig 3C and Fig EV3C). To further demonstrate that chromatin dynamics were associated with fate changes, we directly analyzed the relationship between the two using cells with identical lineage distances. The results showed that a higher chromatin divergence was generally associated with a higher fate divergence, especially between cells at a modest lineage distance (from 6 to 14) (Fig 3D). Thus, chromatin dynamics during lineage progression correlate with lineage-coupled cell differentiation”.

5. I didn't understand their concluding sentence of the first paragraph on p. 12 " Intriguingly, analysis of

lineage-resolved expression data from cells of L1 stage animals (Liu, Long et al., 2009) demonstrated that the lineage effects not only result in gene expression heterogeneity but also are stably maintained after hatching (Appendix Fig S13D)."

Response: Thank you for this comment. Because not all cells at the 350-cell and 600-cell stages have completed terminal cell differentiation, it was unclear whether the observed lineage-dependent cell heterogeneity in chromatin states and gene expression persists in hatched animals, in which most cells have completed differentiation. Therefore, we further analyzed gene expression data at the first larval stage (L1) to examine lineage effects on gene expression, from which we obtained supportive results. We have revised the main text to make this point more clear.

6. In the discussion " Through a multidimensional analysis of the lineage-resolved chromatin landscape, we reveal that the regulation of lineage commitment, anterior-posterior asymmetry, tissue fate specification, cell heterogeneity, and bilateral symmetry establishment are readily inferable from cellular chromatin." implies that the authors have specific information on cellular chromatin state.

Response: We meant that the regulatory processes are readily inferable from the cellular chromatin landscapes. We have revised this sentence as follows for clarity:

"Through a multidimensional analysis of the lineage-resolved chromatin landscape, we found that chromatin dynamics correlate with lineage commitment, anterior-posterior asymmetry, tissue fate specification, cell heterogeneity, and bilateral symmetry establishment. This suggests that regulatory events in cell differentiation could be inferred from cellular chromatin."

7. The legends need much more information in order to understand the figures. e.g, Figure 1G "CAL of

representative embryonic cells. There is no explanation of colors. 1H on histone modifications is impossible to understand without reading the methods. Most figure legends need more information.

Response: We apologize for the confusion. The color gradient represents GFP expression level, which is used to indicate chromatin activity. We have moved all text that describes quality controls from the Methods to the Results section to make the figures in this section easy to follow. Finally, we have significantly expanded all figure legends to provide more context for understanding the results.

8. In the explanation of how GFP was quantified in the methods, the authors say that GFP intensities of the same cell at multiple time points were averaged. How many time points per cell? What is the range?

Response: We have provided information on the time points in the main text and in a figure panel (Fig EV1A).

“On average, GFP expression was measured at 38 consecutive time points (range 13-83) for each traced cell (Fig 1D and Fig EV1A)”

9. Which TADs were used from Crane et al 2015?

Response: As the existence and function of TAD-like structures in *C. elegans* autosomes are currently unclear, we have removed all results related to TADs. Please also see P14, in the response to reviewer # 1.

10. Images in Figure S5 B and C are hard to see. What statistical test was performed when comparing wt and mutant data?

Response: We have enlarged these figure panels (Fig EV2E and F) to better present the results. Cell-by-cell comparisons of GFP expression levels were performed between *wt* and perturbed embryos using the Wilcoxon signed-rank test. We have included this information in the figure legend.

11. Ho et al, 2014 and Evans et al, 2016 derived chromatin states from *C. elegans* embryo histone modification data. Are these states informative with respect to the CAL values? These data may allow the authors to infer the type of chromatin that may be driving the observed relationships.

Response: Thank you for the suggestion. We have performed additional analysis and the results showed that chromatin landscape values are consistent with the classification of chromatin states. We have incorporated these new results into the quality control section of the revised manuscript. However, the relatively small number of reporter integration sites in the genome does not allow us to accurately infer chromatin type directly from reporter expression status.

“Previous studies have also segregated chromatin into various states/domains exhibiting differential activity (Evans *et al*, 2016; Ho *et al.*, 2014). Using the embryonic dataset, we found that average GFP expression levels were significantly higher when located in active chromatin domain/states than when located in silent regions (Fig EV2G and H).”

Reviewer #3:

Major Revision Recommended

In this manuscript, Zhao et al. present a study of position effect variegation (PEV) on reporter gene expression using live imaging during *C. elegans* embryogenesis. The study of PEV on gene expression during differentiation is important, timely, and interesting, and the scope of the study (hundreds of different genomic sites investigated) is technically impressive. The most convincing findings were that individual lineages may have characteristic activity profiles (by their CAL metric), and that tissues arising from diverse lineages have large heterogeneity in CAL divergence among their cells.

Response: We thank the reviewer for these positive remarks.

However, I am not convinced by many of the other conclusions drawn by the authors, which I detail below. Most importantly, the authors should temper their conclusions because this is a reporter-based system with a strong promoter and it is not known if the conclusions will hold true for endogenous sequences. In fact, from Figure S4F, the overall correlation with endogenous gene expression appears poor. This result casts some doubt on the physiological relevance of the reporter data. One practical solution may be to only consider reporter data from an integrated location where there was high correlation with nearby unmodified endogenous genes.

Response: Thank you for the suggestion. We have systematically tempered our conclusions to more accurately summarize the presented data.

The meaning of Fig S4F is two-fold. First, it shows no correlation between GFP and local endogenous gene expression (10-kb interval, $R = 0.13$, $P = 0.24$). This result would be expected because while the expression of endogenous genes is governed by both their promoter

sequence and epigenetic state, a ubiquitous promoter was used to drive the reporter gene, meaning its expression would be regulated by the elements in its own promoter and the epigenetic state around the integration site. This result is consistent with a previous yeast study in which position-effects on the expression of a *kan^R* gene driven by the *TEF* promoter did not correlate with the expression of endogenous genes at the same position (Chen M, Licon K, Otsuka R, Pillus L, Ideker T (2013) Decoupling epigenetic and genetic effects through systematic analysis of gene position. *Cell Rep* 3: 128-137). Furthermore, this finding supports that position-effects on expression of the reporter used in this study are not significantly affected by the local genetic environment (*cis*-elements of endogenous genes). Second, when we extended the interval, we observed a modest but significant correlation (500-kb interval, $R = 0.57$, $P = 3.76E-11$) between expression of GFP and of endogenous genes. Averaging the expression levels of many genes would partially normalize the influence of *cis*-elements on gene expression, allowing the epigenetic effects to stand out. In other words, we used endogenous gene expression over a large genomic interval as proxy for chromatin activity and to assess whether the chromatin landscapes determined from position-effects on GFP expression could indicate chromatin activity. Together with the strong correlation observed between GFP expression and histone modifications, this result suggests the position-effects on reporter expression indicate chromatin states.

Throughout the manuscript, the authors should not assume that chromatin changes precede gene expression and fate changes as the embryo develops, because this has not been proven. They should avoid statements to this end (e.g. chromatin regulation of cell differentiation etc), and instead state their apparent correlations plainly.

Response: Thank you for pointing this out. We have systematically changed the statements so that our conclusions are drawn more accurately.

General comments:

The language used in the title and abstract is vague. These elements should be concise and clearly written, and avoid new unexplained terminology.

Response: We have re-written the title and abstract (the entire manuscript as well) to be more clear. Below please find the new title and abstract.

Chromatin dynamics during *C. elegans* embryogenesis at the single-cell level

Elucidating the chromatin dynamics that orchestrate embryogenesis is a fundamental question in developmental biology. Here, we exploit position-effects on reporter expression as an indicator of chromatin activity and infer the chromatin landscape in every lineaged cell during *C. elegans* early embryogenesis. Systems-level analyses reveal that cellular chromatin is highly informative in defining cellular states and correlates with fate patterning in the early embryos. As cell lineage unfolds, chromatin diversifies in a lineage-dependent manner, with switch-like changes accompanying anterior-posterior fate asymmetry and characteristic chromatin landscapes being established in different cell lineages. Upon tissue differentiation, cellular chromatin from distinct lineages converges according to tissue types but retains stable memories of lineage history, contributing to intra-tissue cell heterogeneity. However, the chromatin landscapes of cells organized in a left-right symmetry pattern are predetermined to be analogous in early progenitors so as to preset equivalent regulatory states. Finally, genome-wide analysis identifies many regions exhibiting concordant chromatin changes that mediate the co-regulation of functionally related genes during lineage differentiation. Collectively, our study reveals the developmental and genomic dynamics of chromatin states at the single-cell level.

Not all figure panels are referenced in the main text, leaving the reader wondering what conclusions can be drawn from these missing panels.

Response: Sorry for this confusion. We have revised the manuscript by placing some text previously in the Methods section in the Results. We hope this modification significantly increases the readability. Moreover, we have carefully checked the manuscript to ensure all figure panels are cited and mentioned in the main text.

Figure legends are under-described, especially in the supplement. They should explain how each experiment was done briefly but clearly and what conclusions can be directly drawn from each.

Response: We have significantly expanded the legends for all figures to provide all essential information.

New terminology like CAL and CDCA which are critical to the message of the manuscript need to be thoroughly and clearly described in the main text.

Response: Thanks for this reminder. We have now added a detailed definition and description of these new terms when they are first presented.

The code used to analyze the data and produce the figures should be deposited in an appropriate online location.

Response: We have provided all code on GitHub (<https://github.com/IGDB-DuLab/Zhao-chromatin>)

Specific comments:

Pg 5 "to explore chromatin regulation of cell differentiation in *C. elegans*," This is very vague, and the data is only correlative, so the authors should be careful not to imply causation without evidence. This wording implies chromatin regulates cell differentiation.

Response: Thank you for pointing this out. We have revised this statement and other related ones.

What they are doing is cataloging the expression levels of a reporter in ~100 different locations using previously generated strains, not measuring cell differentiation. They can say that cells of different lineages and developmental ages have different reporter expression patterns (or chromatin activity landscapes to use the authors' term).

Response: We have changed all statements accordingly.

Pg 5 "These findings will contribute to a systems-level mechanistic understanding of how chromatin regulates cell lineage differentiation in a metazoan embryo across cell lineages, tissue types, and symmetric morphological organization."

I disagree with this statement. The data in the manuscript does not support this claim. There are no proofs in the manuscript of chromatin regulating lineage differentiation.

Response: Thank you for pointing this out. This statement has been replaced with "Our findings contribute to a systems-level understanding of the developmental and genomic dynamics of chromatin at the single-cell level."

Pg 6 "In total, we measured GFP expression in 268 embryos to quantify chromatin activity at 113 genomic positions"

In main text explicitly state how many times each strain was recorded and analyzed. It appears that

only duplicates were performed (from Figure S2). From the duplicate data in Figure S2, the correlation can range from 0 to 1. The authors should discuss this variation in the main text. These experiments form the basis of the entire manuscript and should be conducted in biological triplicates to be convincing. The replicates should be added to Figure S3 so that readers can visually assess the variability.

Response:

(1) In the main text, we have added a discussion on the variability of reporter expression at certain integration sites in certain cells and added a figure (Appendix Fig S2) to illustrate this.

P10 – “While GFP expression was generally consistent between experiment replicates (Fig EV1G), highly variable expression was observed in certain cells at certain genomic positions (Appendix Fig S2). Because only a small number of experimental replicates were performed, we selected four insertion strains exhibiting high variability and quantified GFP expression in more embryos. The correlation of GFP expression between replicates these positions remained low at (Appendix Fig S2), suggesting chromatin states in certain regions could be flexible. Indeed, chromatin state/activity has been previously shown to be variable and stochastic at certain positions (Angermueller *et al*, 2016). Furthermore, when studying the position-effect variegation phenotype of fly eye color, epigenetic silencing of the *white* gene has been shown to be highly stochastic, causing a variegated red and white color phenotype (Timms *et al.*, 2016). ”.

(2) We hope it can be recognized that lineage tracing and curation are very labor-intensive; we thus performed only duplicates in the majority of the cases to balance workload and data reproducibility. Nevertheless, we found that the Pearson correlation coefficient of GFP expression between duplicates is highly comparable to those having a larger number of replicates, suggesting that the estimated reproducibility between replicates is generally accurate.

(3) We have included a 3D visualization of cellular GFP expression for all replicates (Appendix Fig 1) as suggested.

All images of the GFP reporters should have the nuclear mCherry marker side-by-side as a control.

Response: Thanks for the suggestion. We have added the nuclear mCherry images side-by-side in all figure panels showing GFP expression images (Fig 1B, Fig 1C, Fig 5C, Fig EV2E, Fig EV2F).

It is perplexing that reporter expression correlates with histone modification status of the whole embryo (as in mutant and RNAi experiments Figure S5) but not the expression levels of nearby genes (Figure 1I, S4). I do not agree with statements in the text such as "the measured chromatin activity is highly concordant with endogenous gene expression in the same cells" (pg 7) because the data show low Pearson R values 0.3-0.6. It appears that the reporter activity level at its various integrated locations is a poor predictor of endogenous gene activity.

Response: As detailed in our response to another relevant concern raised by this reviewer (P26), the correlation between GFP and endogenous expression is complicated by there being differential *cis*-elements associated with the reporter and endogenous genes. Regarding the correlation coefficient over a large genome interval (500 kb) in single cells, we meant to use the average endogenous gene expression over large genomic regions as proxy for chromatin

activity because in doing so, the effects of differential promoter activities would be normalized. Although the correlation is not very high, it is considerable and significant, making it reasonable to use this result to support that the measured position-effects indicate chromatin activity. We would also like to point out that single-cell gene expression data generated by scRNA-seq tend to be less reliable than bulk cell data, likely affecting the robustness of the correlation.

The term CAL and what data it encompasses should be more clearly stated in the main text.

Response: We have added an explanation of the chromatin activity landscape in the main text.

P7 - “Using GFP expression levels at different genomic positions as a sensor of chromatin activity, we constructed the distribution of chromatin activity across 113 genomic positions (termed the chromatin activity landscape) for all lineage-traced cells (Fig 1F-H and Table EV3). For multiple replicates of GFP expression at the same integration site (range 2-8), only those expressed in more than 60% of replicates were considered as being expressed, and levels were averaged to represent the consensus chromatin activity.”

I am not convinced by the statement that "Given that many embryonic cells have similar developmental fates, this means the cellular CAL is sufficient for defining cellular states at a sub-tissue level." (pg 7)

How does the data in Figure 1L prove this?

Response:

We apologize for this misunderstanding. We have significantly revised this section to better illustrate the point that cellular chromatin landscapes could distinguish cells at the sub-tissue level.

P12 - “We finally examined to what extent the cellular chromatin landscape can distinguish individual cells. We compared the GFP expression patterns across cells and calculated for each pair-wise comparison the number of integration sites at which the GFP expression status is distinct. Intriguingly, for most cell-cell comparisons, the binary GFP expression at many integration sites was distinct, and at a considerable number of integration sites, the expression status can distinguish a cell from many other cells (Fig 2G). On average, cellular chromatin distinguished a cell from 79.4% of other cells if the on/off state of chromatin activity at five or more genomic positions was distinct in a cell-cell comparison. The tissue-level analysis further showed that cellular chromatin not only distinguished a cell from a large fraction of cells belonging to a different tissue type but also distinguished it from a considerable fraction of cells of the same tissue type (Fig 2H). Thus, the cellular chromatin landscape provides rich information for distinguishing cells at a sub-tissue level.”

The authors state that "chromatin transitions predict anterior-posterior asymmetry during lineage progression", however I see no data showing CAL divergence predicts the fate of anterior cells or posterior cells in the embryo.

Response: We meant that when anterior-posterior asymmetric cell divisions occur, chromatin

landscapes in descendant cells tend to exhibit significant differences. We have removed the statement accordingly.

Figure S10 is quite interesting. Can this data (fraction of positions with distinct on/off states) be normalized to lineage distance? This will help to show which results are expected/surprising, and both cases should be discussed.

Response: Thank you for the suggestion. This figure is meant to illustrate that chromatin landscape diversifies across cells in different lineages. While we agree with this reviewer that we could normalize chromatin divergence to the lineage distance between cells and identify unexpected cases, these results will overlap with other findings we mentioned later. Specifically, the tissue-based convergence (Fig 4) and predetermination of similar chromatin landscapes in L-R progenitor cells (Fig 5) are all related to the cases in which lineage distances could not explain the observed chromatin divergences.

The authors should avoid statements such as the underlined: "Collectively, these findings support that cellular chromatin diversifies considerably during early lineage progression and systematically predicts lineage-coupled fate differentiation, including the global lineage-dependent diversification of cell fates, anterior-posterior fate asymmetry, and the establishment of lineage-specific fates." (pg 9) I see no evidence at this point in the manuscript that a particular CAL can predict a particular fate. It is unclear how different CALs are between different lineages. The authors should describe more clearly how CAL divergence is calculated in the main text.

Response: We apologize for the confusion. We have now added a detailed description of how chromatin divergences between cells were measured in the main text. For all chromatin divergence measurements, we used single cells as the basic unit to calculate the Euclidean distance between GFP expression levels across 113 genomic positions, and the divergence

values were averaged accordingly based on what cell groups were being compared. For example, to calculate chromatin divergence between A and B lineages, divergences were first measured between each cell from lineage A and each cell from lineage B, and the values were then averaged to represent the divergence between the two lineages.

The data in Fig S14 is interesting but it is impossible to see the cell IDs in the matrix. This panel should be much bigger. The authors should discuss both unexpected and expected results.

Response: Thank you for the suggestion. We have enlarged this figure panel and provided detailed information on each cell in each cluster in sheet 3 of Table EV7. Furthermore, we added a paragraph to discuss this result.

“While a specific tissue type was enriched in most cell clusters, both pharynx and body wall muscle fates were co-enriched in cells from cluster 3. Interestingly, most of the pharyngeal and muscle cells in this cluster are from the MS lineage (Table EV7), suggesting a lasting influence of lineage effects on tissue differentiation. Although muscle fate was enriched in cells from cluster 4, it explained only a minority (39%) of cell fates. Given that cells in this cluster exhibit very diverse lineage and fate compositions (Table EV7), the implications of a relatively similar chromatin landscape being shared by these seemingly unrelated cells remain to be determined.”

The L-R symmetry section is difficult to understand or find compelling. It is unclear at what point in development CAL divergences are reduced in L-R cell pairs and how this compares to transcriptional divergence.

Response: We have expanded the text to better explain the findings. In short, our results suggest a predetermination of analogous chromatin landscapes in very early progenitors of

future L-R symmetric cells. This is illustrated by *Pnhr-2::GFP* expression in progenitor cells as early as the 50-cell stage (Fig 5G, H, and Appendix Fig S5). Thus, initial chromatin divergences between L-R progenitor cells are suggested to be low. Furthermore, the predetermination model is supported by cellular transcriptome data because when assigning lineage identities to each embryonic cells, most of the L-R progenitor/cells have indistinguishable transcriptomes, and two possible lineage identities (one for the left and another for the right cell) are assigned to the same transcriptome.

Can the authors compare the data derived from the *Pnhr* reporter strains to the *Peef-1A.1* strains? Do close by integration sites produce similar CALs? This could help to show the robustness of CAL as a metric.

Response: Thank you for the suggestion. Unfortunately, only 13 integration sites were obtained for *Pnhr-2::GFP*, and most of them are far (>1 Mb) from the integration sites of *Peef-1A.1::GFP*. This prevented us from accurately testing the correlation of position-effects between the two reporter genes. Nevertheless, in the quality control section (P7-P10, Fig EV2), we have provided multiple lines of evidence to support the validity of using position-effects on *Peef-1A.1::GFP* to indicate chromatin states.

The methodology for CDCA calculations is unclear and therefore the relevance of this data is not apparent. The authors should avoid vague terminology like "inter-cell dynamics".

Response: Sorry for the confusion. We have revised the text and added a figure panel (Fig 6A) to explain the idea better.

“Having examined the cellular dynamics of chromatin activity, we sought to investigate its genomic organization. Specifically, we examined whether concordant changes in chromatin activity across cells predict the functional relevance of genomic regions. We determined to

what extent different genomic regions exhibit similar chromatin changes by calculating divergences of chromatin activity across cells among the 113 genomic positions (Fig 6A), and identified nine clusters of genomic regions (720 pairs) exhibiting similar chromatin changes across cells, which we termed chromatin co-dynamic regions (Fig 6B and Table EV10).”

Thank you for sending us your revised manuscript. We have now heard back from two of the three reviewers who were asked to evaluate your study. Unfortunately, after a series of reminders we did not manage to obtain a report from Reviewer #2. In the interest of time, and since the other two reviewers' recommendations are quite similar, I prefer to make a decision now rather than further delaying the process. You will see from the comments below that the reviewers think that while the majority of the concerns have been addressed, several issues remain. We would therefore ask you to address these issues in the revised manuscript.

On a more editorial level, please address the following issues.

REFEREE REPORTS

Reviewer #1:

Overall, clarification of terms and addition of some more methodology in the text has improved the manuscript. Addition of specific examples to illustrate the point is also helpful. More details and labeling in the figures also helps.

My main continuing concern is regarding my major point. The figure shown in response doesn't really address this point as Euclidean distance often can be a function of differences in magnitude as well as pattern. This can also be true for binarized data due to threshold effects. A similar analysis but using Pearson correlation instead of Euclidean distance would be required to answer the question. I also suggest showing a clustered heatmap of the quantitative measurements (e.g. with cells on one axis and integrants on the other axis).

To illustrate how this might look, I've included a heatmap of expression (as reported in Table EV3) with integrants (rows) clustered and cells (columns) ordered as in the table. From this table I think it is clear that the vast majority of variance in the dataset is differences in the overall intensity of the same underlying pattern (a global scaling effect, e.g. integrant 1 is about twice as bright as integrant 2 in every cell). A very small number of integrants (2-3) show dramatically different patterns, and these integration sites might be quite interesting. It does appear that there may also be some (more subtle) differences in the overall pattern between integrants, but this looks to be on average at least an order of magnitude smaller than the global effect.

One other small point - With regard to "unique identities," - is there evidence for substantial left-right asymmetry between otherwise symmetric cells in this dataset? If not, it would be appropriate to average these to allow more comparisons with the left-right symmetric pairs from the Packer data.

Reviewer #3:

Zhao et al. have made significant efforts to improve the clarity of the text and the data visualization, which has greatly benefitted the manuscript as a whole. I think this study will be of high interest to the chromatin field, as it assesses genomic activity potential in the developing *C. elegans* embryo at the single-cell level using live imaging at an impressive scope. Most of my previous concerns about the robustness and accurate interpretation of the data have been addressed well by the authors and I support publication with some minor revisions (detailed below).

Minor points:

The title and abstract are much improved. However, "chromatin dynamics" is still vague, I would suggest "chromatin activity dynamics" or something along those lines to be more specific, and immediately allow readers to differentiate from chromatin conformational/accessibility dynamics. Throughout the manuscript, "chromatin dynamics" and "chromatin co-dynamics" should also be changed/specified.

Pg.6: It should be stated here in the main text that cell lineages were reconstructed with StarryNite and AceTree with the appropriate citations.

Fig1G, 4C: It would be useful to have the quantification of GFP expression below each image (in the same colormap as 1F) because it is difficult to appreciate by eye the differences between integration sites.

When discussing the correlations between the chromatin activity landscapes/integration sites with epigenetic status, it would be interesting to point out when the authors found unexpected activity landscapes. Which cells have the most divergent landscapes from the population-based modENCODE epigenetic modification datasets? This would exploit a great aspect of their single-cell data - to find something new beyond global correlations (which are already interesting).

Fig 2C,D: What is the meaning of the embryo/circos schematics above the graphs? Do they correspond to a particular bar in the graph? Please explain this in the figure legend.

Fig 2E,F and their description in the main text are vague. The authors are introducing a new metric here since the initial submission, and its importance is not apparent. It is not easy to grasp what "rich information" means in this context or what exactly the IC calculations tell us about the data. I would not be opposed to leaving this paragraph out of the final submission.

When discussing Fig2G,H, the authors should point out and discuss the implications of the unique pattern of intestinal cells, which have very low intra-tissue differences.

Pg. 13 The authors state: "All told, we generated a lineage-resolved chromatin landscape that is dynamic, informative, and biologically relevant in indicating the functional state of chromatin and in defining cellular states." I don't think "defining cellular state" is accurate at this point in the manuscript, "reflecting cell type" or something similar would work.

Pg. 25: State what percentage of co-dynamic regions were within 1Mb vs further apart/different chromosomes.

All figure panels: The number of cells analyzed in each category/bin should be stated in the figure or legend.

Response to reviewers

We thank all reviewers again for the time they spent evaluating our work and for the constructive suggestions made in the interest of improving it. We have revised the text and figures according to their suggestions. Below, we provide point-by-point responses (bold blue) to all concerns and suggestions raised by the reviewers.

Reviewer #1:

Overall, clarification of terms and addition of some more methodology in the text has improved the manuscript. Addition of specific examples to illustrate the point is also helpful. More details and labeling in the figures also helps.

Response: Thank you for these positive comments on the revised manuscript.

My main continuing concern is regarding my major point. The figure shown in response doesn't really address this point as Euclidean distance often can be a function of differences in magnitude as well as pattern. This can also be true for binarized data due to threshold effects. A similar analysis but using Pearson correlation instead of Euclidean distance would be required to answer the question. I also suggest showing a clustered heatmap of the quantitative measurements (e.g. with cells on one axis and integrants on the other axis).

To illustrate how this might look, I've included a heatmap of expression (as reported in Table EV3) with integrants (rows) clustered and cells (columns) ordered as in the table. From this

table I think it is clear that the vast majority of variance in the dataset is differences in the overall intensity of the same underlying pattern (a global scaling effect, e.g. integrant 1 is about twice as bright as integrant 2 in every cell). A very small number of integrants (2-3) show dramatically different patterns, and these integration sites might be quite interesting. It does appear that there may also be some (more subtle) differences in the overall pattern between integrants, but this looks to be on average at least an order of magnitude smaller than the global effect.

Response: Thanks for pointing this out. As requested by this reviewer, we have now provided the results based on calculating Pearson correlation coefficients for both quantitative and binarized expression levels. As shown in Fig 2C and 2D, the correlation coefficient of cellular GFP expression between replicates is significantly higher than between different integrants whether using quantitative (2C) or binarized (2D) expression data. We have also provided a heatmap in Figure 2A comparing GFP expression levels across all integration sites in individual lineaged cells. The text on Pages 11-12 has been modified as well (see below).

Legend: C, D. Distribution of GFP expression similarity (measured as Pearson correlation coefficient) between experimental replicates and between different integrants, calculated using quantitative (C) or binarized (D) expression levels.

Legend: A. Heatmap showing GFP expression levels in individual cells (n = 364, ordered by lineage origin) when integrated into different genomic positions (n = 113, clustered based on expression pattern).

“Having established that the inferred chromatin landscape is reliable, we next determined the extent to which chromatin activity changes across positions and cells (Fig 2A). We first examined whether the expression of *Peef-1A.1::GFP* changes considerably with genomic position, taking the expression variability at each position into account (Fig 2B). For both quantitative and binarized expression, analysis revealed that the correlation of GFP expression between different integration sites was significantly lower than that between replicates at a

given position (Fig 2C and D). Pair-wise comparisons likewise showed that the cellular pattern of GFP expression at a given integration site was, on average, distinct ($R < 0.5$) from 40% (quantitative expression) and 92% (binarized expression) of the patterns resulting from other integration sites. These results suggest that the *eef-1A.1* promoter sequence does not significantly dominate the position effects on GFP expression.”

In addition, we would like to take the opportunity to clarify further why our results support that cellular expression of GFP changes (a readout of chromatin activity) considerably across integrants. First, the promoter used in this study (*eef-1A.1*, also known as *eft-3*) is a well-known strong promoter, which might account for quantitative reduction in expression being more frequently observed than on/off changes. It is reasonable to imagine that a considerable quantitative reduction of chromatin activity assayed by a strong promoter could correspond to more dramatic (on/off) changes in many endogenous contexts. Thus, quantitative reduction of *Peef-1A.1::GFP* could be biologically relevant. Second, while selection of a binary cutoff could be relatively arbitrary, our cutoff for calling expression was carefully determined using the background expression in a strain without the GFP transgene (as described in the Materials and Methods). Thus, a value of 0 in our dataset explicitly indicates that GFP is not expressed. In this regard, the results from calculating the fraction of integrants expressed in each cell (the last paragraph on Page 12) further support that on/off expression changes are frequent. Finally, using the Pearson correlation coefficient (R) as a measurement of similarity, we compared cellular expression (quantitative and binarized) among all integrants and determined that, on average, cellular expression at an integration site is distinct (defined as having an $R < 0.5$)

from 40% (using quantitative data) and 92% (using binarized data) of the cellular expression patterns of other integrants.

Together, we hope the new figure panels and the added clarification could at least partially address the concern raised by this reviewer regarding the level of expression change in individual integrants relative to the global pattern.

One other small point - With regard to "unique identities," - is there evidence for substantial left-right asymmetry between otherwise symmetric cells in this dataset? If not, it would be appropriate to average these to allow more comparisons with the left-right symmetric pairs from the Packer data.

Response: Thank you for this suggestion. In the vast majority of cases, L-R symmetric cells exhibit very similar chromatin activity landscapes. We have now provided an analysis in which the chromatin activity landscapes of L-R symmetric cells were averaged and compared to the single-cell RNA-seq data. This gave out a result similar to the original analysis. This new data is provided in Figure EV 2M.

Reviewer #3:

Zhao et al. have made significant efforts to improve the clarity of the text and the data visualization, which has greatly benefitted the manuscript as a whole. I think this study will be of high interest to the chromatin field, as it assesses genomic activity potential in the developing *C. elegans* embryo at the single-cell level using live imaging at an impressive scope. Most of my previous concerns about the robustness and accurate interpretation of the data have been addressed well by the authors and I support publication with some minor revisions (detailed below).

Response: We thank this reviewer for the kind words and for supporting the publication of our manuscript.

Minor points:

The title and abstract are much improved. However, "chromatin dynamics" is still vague, I would suggest "chromatin activity dynamics" or something along those lines to be more specific, and immediately allow readers to differentiate from chromatin conformational/accessibility dynamics. Throughout the manuscript, "chromatin dynamics" and "chromatin co-dynamics" should also be changed/specified.

Response: Thank you for the suggestion. We have now changed "chromatin dynamics" to "chromatin activity dynamics", "chromatin divergence" to "chromatin activity divergence", and "chromatin co-dynamics" to "chromatin activity co-dynamics" throughout the manuscript. Furthermore, to better highlight the data and findings of this work, we have changed the title to

“Single-cell dynamics of chromatin activity during cell lineage differentiation in *C. elegans* embryos”.

Pg.6: It should be stated here in the main text that cell lineages were reconstructed with StarryNite and AceTree with the appropriate citations.

Response: We have now mentioned the software and added corresponding citations in the main text.

“Cell lineages were reconstructed based the automatic identification and tracing of all cells via the mCherry signal using StarryNite and AceTree software (Bao et al, 2006; Boyle et al, 2006; Katzman et al, 2018; Santella et al, 2014; Santella et al, 2010), followed by multiple rounds of manual curation.”

Fig1G, 4C: It would be useful to have the quantification of GFP expression below each image (in the same colormap as 1F) because it is difficult to appreciate by eye the differences between integration sites.

Response: Thank you for the suggestion. Colormaps are now added below corresponding micrographs to show quantitative differences in GFP expression level.

When discussing the correlations between the chromatin activity landscapes/integration sites with epigenetic status, it would be interesting to point out when the authors found unexpected activity landscapes. Which cells have the most divergent landscapes from the population-based modENCODE epigenetic modification datasets? This would exploit a great aspect of their single-cell data - to find something new beyond global correlations (which are already interesting).

Response: Thank you for the comments. We have now discussed this point in the 2nd paragraph on Page 14 and provided new figure panels in Fig 3 (I-K).

“Finally, the cellular resolution of our data enabled identifying potential cell- and position-specific chromatin activity that is unable to be obtained from the bulk data of cell populations. Although the chromatin activity described here is globally concordant with previously-defined chromatin regions for which a silent/active state is evident in bulk epigenomic data (Fig EV2G and H), this activity exhibited considerable cell-specificity at given genomic positions. For example, certain integrants located in regions with a global silent/active state exhibited divergent activity in specific cells (Fig 2I). Moreover, certain cells also exhibited chromatin activity landscapes that diverged from those predicted by cell population-based histone modification datasets (Ho et al., 2014). Correlation analysis comparing the observed chromatin

activity landscape in each cell with that predicted by a combination of 19 types of histone modifications revealed that in certain cells, the predictive power of bulk histone modifications was low (Fig 2J). Interestingly, these cells were significantly enriched for neuronal cells from the ABal lineage (Fig 2K, 3.36-fold enrichment, $P = 4.22E-21$, Fisher's exact test). Thus, a lineage-resolved single-cell dataset provides the opportunity to pinpoint genomic regions and cells having distinct chromatin activity.”

Legend:

I. Heatmap showing GFP expression levels in individual cells ($n = 364$) when integrated into previously-defined genomic regions having global silent or active chromatin state (Ho *et al.*, 2014). On the right are indicated four representative integration sites at which GFP expression is distinct from the global pattern.

J. Distribution of Pearson correlation coefficients comparing the observed chromatin activity landscapes in individual cells to those predicted by a combination of 19 types of histone modifications ($n = 364$).

K. Correlation of cellular chromatin activity landscape to the global landscape predicted by histone modifications, contextualized by cell lineage origin and tissue fate. On the lineage tree, traced terminal cells are color-coded according to the correlation coefficient between observed

and predicted chromatin activity; the bottom 25% (blue) are considered divergent. The barcode below the cell lineage indicates the tissue fate of each cell.

Fig 2C,D: What is the meaning of the embryo/circos schematics above the graphs? Do they correspond to a particular bar in the graph? Please explain this in the figure legend.

Response: These schematics are meant to provide representative examples showing the positional and cellular dynamics of chromatin activity. We have now added a description in the figure legend.

“E. Distribution of the proportion of cells that express GFP at each integration site. Schematics on the top show expressing cells (green) of representative integrants.

F. Distribution of the proportion of integration sites at which GFP is expressed in each cell. Schematics on the top show the expressed integrants (green) in representative cells.”

Fig 2E,F and their description in the main text are vague. The authors are introducing a new metric here since the initial submission, and its importance is not apparent. It is not easy to grasp what "rich information" means in this context or what exactly the IC calculations tell us about the data. I would not be opposed to leaving this paragraph out of the final submission.

Response: Thank you for pointing this out. We have removed these figure panels and associated text for clarity.

When discussing Fig2G,H, the authors should point out and discuss the implications of the unique pattern of intestinal cells, which have very low intra-tissue differences.

Response: Thank you for pointing this out. We now discuss this as follows:

“The only exception was the intestine cells, for which distinguishing chromatin activity occurred at only a very few integration sites. One possibility is that intestine cells exhibit limited functional diversification due to being derived clonally from a single progenitor cell (called E) and all intestine cells or their progenitors uniformly expressing most master regulators of intestine differentiation (Maduro & Rothman, 2002; McGhee, 2007).”

Pg. 13 The authors state: "All told, we generated a lineage-resolved chromatin landscape that is dynamic, informative, and biologically relevant in indicating the functional state of chromatin and in defining cellular states." I don't think "defining cellular state" is accurate at this point in the manuscript, "reflecting cell type" or something similar would work.

Response: Thank you for the suggestion. We have changed the word “define” to “distinguish” throughout the manuscript.

Pg. 25: State what percentage of co-dynamic regions were within 1Mb vs further apart/different chromosomes.

Response: We have now included the percentage in the main text.

“A small fraction (4%) of chromatin activity co-dynamic regions were linked on the chromosome at a distance of less than one Mb; most (80%) co-dynamic regions were located on different chromosomes.”

All figure panels: The number of cells analyzed in each category/bin should be stated in the figure or legend.

Response: Thank you for pointing this out. We have now provided detailed information on cell numbers in figure legends.

Thank you for sending us your revised manuscript. We have now heard back from the reviewer who was asked to evaluate your study. As you will see the reviewer is satisfied with the modifications made and thinks that the study is now suitable for publication.

Before we can formally accept your manuscript, we would ask you to address the remaining minor issues raised by the reviewer.

On a more editorial level -

REFEREE REPORTS

Reviewer #1:

I thank the authors for the thoughtful response to my major concern. I'm satisfied with the current version although suggest a small textual change:

- 1) The most compelling argument in the authors' more detailed response is that quantitative changes are likely meaningful (especially given that they are consistent between replicates). I agree with this and suggest that a concise version of this description be added either immediately before or after paragraph 1 of the section "Cellular chromatin activity is dynamic and informative for distinguishing cellular states"
- 2) This is especially important because the other point made ("our cutoff for calling expression was carefully determined using the background expression in a strain without the GFP transgene...Thus, a value of 0 in our dataset explicitly indicates that GFP is not expressed." Isn't quite valid. As I understand this procedure, the threshold is conservative, so a value of 1 'explicitly indicates that GFP is expressed' but a zero value could indicate sub-detection expression. Indeed all live imaging microscopy platforms have a nonzero true expression level that cannot be detected. Consistent with this, most of the cells that have many "Zero" values in Figure 2A also have low quantitative levels in other integrants. For this reason I think that the quantitative changes are a stronger argument than the binary changes.

Response to reviewer

We thank Reviewer # 1 for the further constructive suggestions. Below, we provide point-by-point responses (blue) to all comments made by Reviewer # 1.

Reviewer #1:

I thank the authors for the thoughtful response to my major concern. I'm satisfied with the current version although suggest a small textual change:

1) The most compelling argument in the authors' more detailed response is that quantitative changes are likely meaningful (especially given that they are consistent between replicates). I agree with this and suggest that a concise version of this description be added either immediately before or after paragraph 1 of the section "Cellular chromatin activity is dynamic and informative for distinguishing cellular states"

Response: Thank you for the suggestion. We have added a couple of sentences at the end of paragraph 1 to illustrate that quantitative changes are likely meaningful: "It should be noted that the promoter used in this study is a well-known strong promoter, which might account for quantitative reduction in expression being more frequently observed than on/off changes. Thus, a considerable quantitative reduction of chromatin activity assayed here could correspond to more dramatic (on/off) changes in many endogenous contexts."

2) This is especially important because the other point made ("our cutoff for calling expression was carefully determined using the background expression in a strain without the GFP transgene...Thus, a value of 0 in our dataset explicitly indicates that GFP is not expressed." Isn't quite valid. As I understand this procedure, the threshold is conservative, so a value of 1 'explicitly indicates that GFP is expressed' but a zero value could indicate sub-detection expression. Indeed all live imaging microscopy platforms have a nonzero true expression level that cannot be detected. Consistent with this, most of the cells that have many "Zero" values in Figure 2A also have low quantitative levels in other integrants. For this reason I think that the quantitative changes are a stronger argument than the binary changes.

Response: Thank you for clarifying this point. We have removed the text "It should be noted that the binary cutoff for calling expression was carefully determined by comparing the cellular GFP intensity in each integrant to the background intensity in a strain without the GFP transgene (as described in Materials and Methods)" from the manuscript.

Thank you again for sending us your revised manuscript. We are now satisfied with the modifications made and I am pleased to inform you that your paper has been accepted for publication.

Corresponding Author Name: Zhuo Du

Manuscript Number: MSB-2020-10075